| 1  | TECO-CNP Sv1.0: A coupled carbon-nitrogen-phosphorus model with data                                                                                          |
|----|---------------------------------------------------------------------------------------------------------------------------------------------------------------|
| 2  | assimilation for subtropical forests                                                                                                                          |
| 3  | Fangxiu Wan <sup>1</sup> , Chenyu Bian <sup>1</sup> , Ensheng Weng <sup>2</sup> , Yiqi Luo <sup>3</sup> , Kun Huang <sup>1</sup> , Jianyang Xia <sup>1*</sup> |
| 4  | <sup>1</sup> Zhejiang Tiantong Forest Ecosystem National Observation and Research Station,                                                                    |
| 5  | Research Center for Global Change and Ecological Forecasting, School of Ecological and                                                                        |
| 6  | Environmental Sciences, Institute of Eco-Chongming, East China Normal University,                                                                             |
| 7  | Shanghai, China.                                                                                                                                              |
| 8  | <sup>2</sup> Center for Climate Systems Research, Columbia University, New York, USA                                                                          |
| 9  | <sup>3</sup> School of Integrative Plant Science, College of Agriculture and Life Sciences, Cornell                                                           |
| 10 | University, Ithaca, NY 14853, USA.                                                                                                                            |
| 11 | *Corresponding to: Jianyang Xia (jyxia@des.ecnu.edu.cn)                                                                                                       |
| 12 |                                                                                                                                                               |
| 13 |                                                                                                                                                               |

#### 14 Abstract

Subtropical forests play a crucial role in the global carbon cycle, yet their carbon sink capacity is significantly constrained by phosphorus availability. Models that omit phosphorus dynamics risk overestimating carbon sinks, potentially undermining the scientific basis for carbon neutrality strategies. In this study, we developed TECO-CNP Sv1.0, a coupled carbon-nitrogen-phosphorus model based on the Terrestrial ECOsystem (TECO) model, which explicitly captures key biogeochemical interactions and nutrientregulated carbon cycling. The model simulates how plant growth and carbon partitioning respond to both external soil nutrient availability and internal physiological constraints, enabling plant acclimation to varying nutrient conditions. Using observations from a phosphorus-limited subtropical forest in East China, we first evaluated the model's performance in estimating state variables with empirically calibrated parameters. Compared to the C-only and coupled C-N configurations, the CNP model more accurately reproduced the observed pools of plant and soil C, N, and P. To systematically optimize model parameters and reduce uncertainties in predictions, we further incorporated a builtin data assimilation framework for parameter optimization. The CNP model with optimized parameters significantly improved carbon flux estimates, reducing root mean square errors and enhancing concordance correlation coefficients for gross primary productivity, ecosystem respiration, and net ecosystem exchange. By explicitly incorporating phosphorus dynamics and data assimilation, this study provides a more accurate and robust framework for predicting carbon sequestration in phosphorus-limited subtropical forests.

#### Introduction

35

36

Accurately representing phosphorus (P) cycling in land surface models (LSMs) is crucial 37 for projecting terrestrial carbon sink dynamics under climate change (Wieder et al., 2015). 38 As an essential element, P availability regulates plant growth and ecosystem productivity 39 (Walton et al., 2023; Vitousek et al., 2010). For instance, nutrient addition experiments in 40 an old-growth Amazon rainforest demonstrated that net primary productivity increased 41 exclusively with P addition (Cunha et al., 2022). Likewise, in subtropical mature forests, 42 soil P availability was found to exert dominant control over plant functional traits at both 43 species and community levels (Cui et al., 2022). Recent global syntheses have revealed a 44 more widespread distribution of terrestrial P limitation than previously recognized (Hou et 45 al., 2020; Du et al., 2020; Xia & Wan, 2008; Elser et al., 2007). More concerning is that P 46 limitation is expected to intensify (Wang et al., 2023; Luo et al., 2022) due to factors such 47 as N deposition-induced N:P stoichiometric imbalance (Peng et al., 2019; Lu and Tian, 48 2017; Du et al., 2016; Peñuelas, 2013) and reduced P availability under elevated CO<sub>2</sub> 49 concentration (Wang et al., 2023). Consequently, incorporating P limitation into LSMs has 50 become a pressing challenge for improving carbon cycle projections (Fisher & Koven, 51 2020; Achat et al., 2016; Reed et al., 2015). 52 To address this challenge, several modeling groups have incorporated a prognostic P 53 cycle into their existing frameworks over the past decade, including CASACNP (Carnegie-54 Ames-Stanford Approach; Wang et al., 2010), JSBACH (Jena Scheme for Biosphere-55 Atmosphere Coupling in Hamburg; Goll et al., 2012), CLM-CNP (Community Land 56 Model; Yang et al., 2014), among others. These pioneering efforts in coupled carbon-57 nitrogen-phosphorus (C-N-P) modeling have laid a solid foundation for increasing incorporation of P cycling in LSMs (e.g., Goll et al., 2017; Nakhavali et al., 2022) and 58 59 demographic vegetation models (Knox et al., 2024), shedding light on how P limitation 60 constrains ecosystem productivity under elevated atmospheric CO<sub>2</sub> (Wang et al., 2024; Fleischer et al., 2019; Medley et al., 2016). However, current C-N-P models often yield 61 62 "right answers for wrong reasons" (Jiang et al., 2024a), largely due to two key limitations: 63 (1) calibration and validation data are predominantly derived from a narrow range of 64 ecosystems, with most coupled C-N-P models relying on *in-situ* data from tropical regions,

particularly Hawaii and the Amazon (e.g., Nakhavali et al., 2023; Yang et al., 2014; Goll et al., 2012, 2017; Zhu et al., 2016), and (2) oversimplified representations of P cycling processes (Achat et al., 2016; Reed et al., 2015), such as the absence of physiological mechanisms governing vegetation P uptake (Jiang et al., 2019). Addressing these gaps requires advancing the coupled C-N-P model with improved mechanistic process-based representations and broader ecosystem applicability (Jiang et al., 2024a).

Subtropical forest ecosystems are recognized as important carbon sinks in the global carbon cycle (Pan et al., 2024; Keenan et al., 2018; Yu et al., 2014). In particular, East Asian monsoon subtropical forests exhibit high carbon sink capacity, with an average net ecosystem productivity of about 400 g C m<sup>-2</sup> yr<sup>-1</sup> (Yu et al., 2014). These ecosystems are likely subject to substantial phosphorus limitation, as evidenced by a meta-analysis of nutrient addition experiments showing that forest productivity exhibits the strongest standardized response to P addition in the subtropical regions (25-40 latitude; Hou et al., 2021). Moreover, intensive nitrogen deposition may further exacerbate P limitation (Zhu et al., 2016; Yu et al., 2014). Accurately projecting the future carbon sink capacity of subtropical forests is crucial for assessing their role in climate change mitigation (Friedlingstein et al., 2023; Requena-Suarez et al., 2019; Grassi et al., 2017). However, substantial uncertainties remain in current model projections of subtropical carbon dynamics (Wei et al., 2024), highlighting the urgent need for improved carbon cycle predictions through better representation of coupled C-N-P interactions in these regions.

In this study, we develop TECO-CNP Sv1.0, an advanced version of the Terrestrial ECOsystem (TECO) model (Weng & Luo, 2008, 2011), incorporating detailed mechanistic representations of coupled C-N-P cycling processes, such as dynamic plant growth response to soil available nutrient through modified growth rates and allocation patterns, and the combined physical and physiological controls on phosphorus uptake. Additionally, we integrated a data assimilation module based on a Bayesian probabilistic inversion approach (Xu et al., 2006; Ma et al., 2017; Shi et al., 2016, 2018; Zhou et al., 2020), providing an efficient framework for model reparameterization and broader applications. Based on comprehensive observations from a P-limited subtropical evergreen broadleaf forest in eastern China, we further test two key hypotheses: (1) the CNP model can

- reproduce ecosystem state variables through traditional spin-up and manual parameter
- tuning, and (2) the built-in data assimilation system can substantially improve carbon flux
- predictions.

#### 2 Materials and Methods

#### 2.1 TECO developments

- The TECO-CNP model has evolved from its precursor, the Terrestrial Ecosystem model
- (TECO, Weng & Luo, 2008). The TECO model is a process-based ecosystem model
- encompassing eight organic carbon pools and a plant non-structural carbohydrate (NSC)
- pool (Weng & Luo, 2008). The representation of the NSC pool in TECO is advantageous
- for capturing the seasonal decoupling of growth and nutrient acquisition within plants
- (Zavisic & Polle, 2018; Jones et al., 2020) and for managing carbon that is not utilized for
- plant growth under nutrient-limited conditions (Nakhavali et al., 2022; Haverd et al., 2018).
- The TECO model has been part of model intercomparison ensembles (Zaehle et al., 2014;
- De Kauwe et al., 2014) and has been applied across diverse ecosystem types, such as
- grassland (Weng & Luo, 2008; Zhou et al., 2021), temperate coniferous forests (Luo et al.,
- 2003; Weng & Luo, 2011; Jiang et al., 2017) and deciduous broadleaf forests (Jiang et al.,
- 2017) and northern peatland (Ma et al., 2017, 2022; Huang et al., 2017).
- Simplified N and P cycling were incorporated in the TECO successively (Shi et al.,
- 2016; Du et al., 2018; Du et al., 2021), where the structure of the carbon processes was
- expressed as a matrix form (Luo et al., 2003; Xu et al., 2006; Weng & Luo, 2011). Thus,
- the photosynthesis was simulated aided by an external model; for instance, Shi et al. (2016)
- utilized MAESTRA to generate the gross primary productivity. The processes related to
- the N and P cycle were only represented in a parsimonious way in the matrix versions. For
- example, the nutrient uptake process was simplified at a constant rate, and the interactions
- of carbon, nitrogen, and phosphorus were treated implicitly (Shi et al., 2016; Du et al.,
- 2021).
- In this study, we developed TECO-CNP, a coupled C-N-P model based on the full
- version of TECO, which fundamentally differs from previous matrix-based approaches.
- This new model explicitly represents the mechanistic processes of nutrient cycling (Sect.

2.2), with a focus on the regulation of carbon cycling by nutrients. Specifically, the model incorporates four key nutrient-carbon interactions: (1) growth rate limitations controlled by internal plant nutrient concentrations and nutrient supply-demand relationships; (2) allocation patterns dependent on nitrogen and phosphorus availability; (3) decomposition processes constrained by microbially-mediated nutrient availability; and (4) carbon costs associated with nutrient uptake and fixation. These process-based implementations, which aim to provide a more realistic representation of terrestrial biogeochemical cycles, are described in detail in the following sections.

# 2.2 Model description

We introduce a comprehensive biogeochemical N and P cycle into the full TECO, named TECO-CNP Sv1.0. Key processes of N and P cycling and their interactions with the carbon cycle have been represented using reliable mechanistic assumptions based on our experimental measurements or validated by state-of-the-art LSMs. In the following sections, we first document an overview of the carbon cycle and highlight the effects of nutrient limitation on the carbon cycle in Sect. 2.2.1. We then describe the shared and specific N and P cycling processes in Sects. 2.2.2 and 2.2.3, respectively.

# 2.2.1 Nutrient-limited carbon cycle

The carbon cycle in the new model builds upon the TECO model, incorporating processes such as photosynthesis, plant growth controlled by allocation and phenology, autotrophic and heterotrophic respiration, litter production, and carbon transfer (Fig. 1). See Luo et al. (2003) and Weng & Luo (2008) for detailed descriptions. These processes regulate the dynamics of plant, litter, and soil pools (Fig. 2). Nutrients directly or indirectly constrain them. For instance, plant growth rates and carbon allocation strategies are directly influenced by internal nutrient availability within pools and the availability of soil-accessible nitrogen and phosphorus. Additionally, resource limitations adhere to Liebig's law of the minimum, where the nutrient-constrained process is hindered only by the most limiting resources (Rastetter, 2011).

Figure 1. The schematic diagram of the biogeochemical processes of the carbon, nitrogen, and phosphorus cycles and associated interactions in TECO-CNP. Representation of carbon cycling processes controlled by nitrogen and phosphorus in TECO-CNP. Solid lines indicate carbon cycling processes (labelled 1-7) comprise (1) photosynthesis, (2) carbon allocation, (3) plant growth, (4) autotrophic respiration, (5) litter production, (6) carbon transfer, and (7) heterotrophic respiration. These processes are controlled directly by nitrogen and phosphorus (black control characters) or indirectly (colorless control characters). Dashed lines indicate the common processes that control the dynamics of soil-available nitrogen and phosphorus, simplified as plant uptake, mineralization, immobilization, biogeochemical mineralization, and external input and loss. Irregular pink shapes represent competition for soil available nitrogen and phosphorus between plants and microorganisms. Min., mineralization; BMin., biochemical mineralization; Imm., Immobilization.

**Figure 2. Model structure of TECO-CNP.** The model represents the nine organic carbon, nitrogen, and phosphorus stocks within the plant (denoted as Q1-4), litter (Q5-7), and soil (Q7-9). Fluxes among these organic pools are depicted by black arrows. Specific N and P fluxes are indicated by dark red arrows, with associated processes labeled accordingly. Min denotes mineralization, and Imm denotes immobilization. The circled numbers (1-7) correspond to the carbon cycling processes in Fig. 1.

The canopy-level photosynthesis is simulated using a two-leaf model, which consists of a radiation sub-model and a coupled sub-model of stomata, photosynthesis, and transpiration for both sunlight and shaded leaves (Wang & Leuning, 1998). Leaf photosynthesis is estimated by the equations derived from the Farquhar model (Farquhar et al., 1980) and a stomatal conductance model (Ball et al., 1987; Leuning et al., 1995). The photosynthesis of a single leaf is then scaled up to the canopy level (Wang & Leuning, 1998). We hypothesize that plant photosynthesis is downregulated as photosynthetic surface area decreases when nutrient limits plant growth. Plant growth is adjusted based on the nutrient limitation factor calculated at each time step, meaning that plants tend to reduce growth under low nutrient conditions to avoid nutrient deficiency within the organism (Veneklaas et al., 2012). Accordingly, the nutrient-constrained growth rate ( $GP_a$ ) is dependent on the potential growth rate ( $GP_p$ ) and nutrient limitation scalar for plant growth ( $L_{GP}$ ) as the following equation:

$$GP_{a,i} = GP_{p,i} * L_{GP} , \qquad (1)$$

where subscript i indicates leaf (i = I), wood (i = 2), root (i = 3) or reproduction (i = 4) (Table 1). The difference between actual and potential plant growth is referred to as excess carbon, which implicitly represents the carbon lost from the NSC pool through various pathways to cope with nutrient limitations.

The nutrient limitation scalar for plant growth incorporates both the nutrient status of plant tissues and soil nutrient supply (Fig. 1b). which can be expressed as:

$$L_{GP} = L_{in,leaf} L_{sp} , \qquad (2)$$

where  $L_{in,leaf}$  and  $L_{sp}$  represent the nutrient limitation factors derived from leaf nutrient concentration (Eqs. 3-5) and the nutrient demand-supply process (Eqs. 6-8), respectively. Shifts in leaf nutrient concentrations act as a potential limiting factor for plant growth, implying the mechanism by which changes in leaf nutrient concentration can impact photosynthesis (Ellsworth et al., 2022; Sterner & Elser, 2002). Description of limitation factors that account for plant tissue's nutrient concentration can be given by:

$$L_{in,i} = \min(L_{in,N,i}, L_{in,P,i}), \qquad (3)$$

$$L_{in,N,i} = \frac{R_{N,i}}{R_{N,i} + k_{CN}},\tag{4}$$

$$L_{in,P,i} = \frac{R_{P,i}}{R_{P,i} + k_{CP}},\tag{5}$$

where  $R_N$  and  $R_P$  represent the C:N ratios and C:P ratios, respectively.  $k_{CN}$  and  $k_{CP}$  are empirical parameters. A study by Cui et al. (2020) reveals that the Tiantong site is identified as a P-limited ecosystem, as indicated by the leaf N:P thresholds from Koerselman and Meuleman (1996). Thus, we adopted the values of  $k_{CP}$  (0.0006 gP gC<sup>-1</sup>) in Wang et al. (2010) to achieve N limitation when N:P < 16 (gN gP<sup>-1</sup>), and otherwise, plant growth is limited by P.  $k_{CN}$  (0.01 gN gC<sup>-1</sup>) is given based on the results of Linder & Rook (1984).

**Table 1.** Variables for carbon cycling processes in TECO-CNP.

| Variables          | les Description                                                                                                            |                                    |
|--------------------|----------------------------------------------------------------------------------------------------------------------------|------------------------------------|
| GPp                | Potential plant growth rate without nutrient limitation                                                                    | gC m <sup>-2</sup> h <sup>-1</sup> |
| $GP_a$             | Nutrient-limited plant growth rate                                                                                         | $gC m^{-2} h^{-1}$                 |
| $D_{a,x}$          | Actual decomposition rate of litter pool $m$ or soil pool $j$ , accounting for nutrient limitation, $x = m, j$             | $gC m^{-2} h^{-1}$                 |
| $D_x$              | Potential decomposition rate of litter pool $m$ or soil pool $j$ , controlled by soil temperature and moisture, $x = m, j$ | $gC m^{-2} h^{-1}$                 |
| $NPP_i$            | Net primary productivity allocated to plant pool i                                                                         | $gC m^{-2} h^{-1}$                 |
| $F_{new,C,i} \\$   | Newly input carbon from NSC pool for plant growth                                                                          | $gC m^{-2} h^{-1}$                 |
| $b_{C,i}$          | Allocation fraction of carbon to plant pool i                                                                              | unitless                           |
| $r_{i,j}$          | Fraction of carbon from plant pool $i$ to litter pools $j$                                                                 | unitless                           |
| $BM_{root}$        | Plant root biomass                                                                                                         | g biomass m <sup>-2</sup>          |
| $BM_{root^{\ast}}$ | Root biomass density                                                                                                       | g biomass m <sup>-3</sup>          |
| $f_{nsc}$          | Plant labile carbon limiting factor                                                                                        | unitless                           |
| $f_{\mathrm{W}}$   | Soil moisture limiting factor                                                                                              | unitless                           |
| $f_T$              | Soil temperature limiting factor                                                                                           | unitless                           |
| W                  | Soil water availability index                                                                                              | unitless                           |
| κ                  | Light availability factor                                                                                                  | unitless                           |

\* *i* indicates leaf (i = 1), wood (i = 2), root (i = 3) or reproduction (i = 4), *j* indicates metabolic litter (j = 2) or structure litter (j = 6), and *m* indicates fast SOM (m = 7), slow SOM (m = 8) and passive SOM (m = 9).

The nutrient demand-supply limitation factor is calculated as a function of plant nutrient uptake and demand. When nutrient demand is not satisfied, the value of the limitation factor falls below one, thereby impacting plant growth. This assumption aligns with field findings that reveal an increase in plant productivity following nutrient addition (Cunha et al., 2022; Liang et al., 2021). Description of nutrient demand-supply limitation factor ( $L_{sp}$ ) can be given by:

$$L_{sp} = \min(L_{sp,N}, L_{sp,P}), \qquad (6)$$

$$L_{sp,N} = \frac{1}{1 + \exp\left(-12 * \frac{F_{up,N}}{F_{dm,N}} + 6\right)},$$
(7)

$$L_{sp,P} = \frac{1}{1 + \exp\left(-12 * \frac{F_{up,P}}{F_{dm,P}} + 6\right)},$$
(8)

where  $F_{up,N}$  and  $F_{up,P}$  represent plant nutrient uptake for N and P, respectively, which is determined by both supply and demand (Eq. 23).  $F_{dm,N}$  and  $F_{dm,P}$  represent the plant required N and P to sustain a given NPP (Eq. 24). We implemented a logistic function to represent the phosphorus limitation factor, which provides a more mechanistically sound representation of nutrient limitation compared to the simple linear ratio. This formulation ensures a smooth transition between phosphorus-limited and phosphorus-sufficient conditions, with values bounded between zero and one. The coefficients were carefully selected to maintain appropriate sensitivity in the transition zone while avoiding unrealistic sharp thresholds. This sigmoidal response more accurately reflects the gradual physiological adjustments of plants to varying nutrient availability. It is consistent with a theoretical understanding of the effects of nutrient limitation on plant growth. The method of determining whether plants are nutrient-limited based on the supply-demand method is widely employed in many models, for example, CASACNP (Wang et al., 2010), CLM-CNP (Yang et al., 2014), and ORCHIDEE (revision 4520; Goll et al., 2017).

The carbohydrates available for plant growth will be redistributed among the plant pools based on their actual growth rates. A prescribed proportion of those allocated to reproductive processes (Sitch et al., 2003; Smith et al., 2001), such as flower formation, fruit development, and seed production, is stored in the reproductive pool. Vegetation

growth is assumed to take priority over reproduction (Zust et al., 2015; Tang et al., 2021). Thus, the plant's reproductive allocation is zero when the leaf area index (LAI) is below the minimum threshold. When LAI exceeds the minimum threshold, 12% of the available carbon is allocated to the reproduction pool. The remaining carbon is subsequently distributed among leaf, wood, and root pools based on a resource limitation allocation scheme.

The dynamic allocation for leaf, wood, and root is regulated by light availability, soil water supply, canopy phenological status (Luo et al., 1995; Denison & Loomis, 1989; Arora and Bore, 2005), and plant's internal nutrient status (Fig. 1b). This allocation strategy permits a reduction in photosynthetic surface area and enhanced root growth under nutrient limitation, exemplifying a structural adjustment in line with the observations (Keith et al., 1997; Thomas et al., 2015; Yan et al., 2016). The allocation fractions for leaf, wood, and root are given by:

$$b_{C,leaf} = \frac{\varepsilon_L * L_{in,leaf}}{1 + \omega(2 - \kappa - W)}, \tag{9}$$

$$b_{C,wood} = \frac{\varepsilon_w * L_{in,wood} + \omega(1 - \kappa)}{1 + \omega(2 - \kappa - W)},$$
(10)

$$b_{C,root} = \frac{(1 - \varepsilon_L * L_{in,leaf} - \varepsilon_W * L_{in,wood}) + \omega(1 - \kappa)}{1 + \omega(2 - \kappa - W)} = 1 - b_{C,leaf} - b_{C,wood},$$
(11)

where  $b_{C,leaf}$ ,  $b_{C,wood}$  and  $b_{C,root}$  represent the carbon fractions available for growth allocated to leaf, wood, and root, respectively. W is the root zone soil water availability stress factor (Arora & Boer, 2005). The soil water availability is weighted by the existing fraction of roots in each soil layer (Weng & Luo et al., 2008; Arora & Boer, 2005).  $\kappa$  represents the availability of light (Arora & Boer, 2005). Parameters  $\varepsilon_w$ ,  $\varepsilon_L$ , and  $\omega$  are calibrated based on the broadleaf evergreen PFT parameters given in Arora and Boer (2005).  $L_{in,wood}$  and  $L_{in,leaf}$  represent the limitation factor determined by the nutrient status of tissues (Eqs. 3-5), designed to capture the reduction of carbon allocated to leaf and wood as an adaptation to nutrient limitation (Binkley et al., 1995; Yan et al., 2016) and the negative correlation between fine root biomass and soil fertility (Fortier et al., 2019).

Canopy phenology is represented by annual variation in LAI. The beginning of a growing season is determined by growing degree days. Leaf senescence is triggered by low air temperatures and soil moisture (Arora & Boer, 2005), resulting in a reduction of LAI. The litter production rates of wood and roots are prescribed. The phenological parameters are adjusted according to the vegetation characteristics in the studied evergreen forest (Table S1).

Carbon transfer between litter pools and soil organic pools through microbial decomposition (Luo & Reynold, 1999; Weng & Luo, 2008). The decomposition of litter and soil organic matter (SOM) is diminished when the amount of available inorganic N and P restricts nutrient immobilization during decomposition:

$$D_{a,j} = D_j * L_{de} , \qquad (12)$$

$$D_{am} = D_m * L_{de} , (13)$$

where j indicates metabolic litter (j = 5) or structure litter (j = 6), and m indicates fast SOM (m = 7), slow SOM (m = 8) and passive SOM (m = 9).  $D_a$  is the nutrient-constrained decomposition rate, and D is the default decomposition rate controlled by the soil temperature and moisture (Weng & Luo, 2008).  $L_{de}$  is the limiting factor of decomposition, and the calculation involves dividing the un-limited net mineralization rate by the size of the inorganic nutrient pool, which can be addressed in the following equations:

$$L_{de,N} = \max\left(0, 1 + \frac{F_{N,net}'}{N_{min}}\right),$$
 (14)

$$L_{de,P} = \max\left(0, 1 + \frac{F_{P,net}'}{P_{lab}}\right),$$
 (15)

$$L_{de} = \min(L_{de,N}, L_{de,P}), \tag{16}$$

where  $F'_{N,net}$  and  $F'_{P,net}$  represent the net mineralization rate for nitrogen and phosphorus, 274 respectively, assuming no nutrient limitation on mineralization (Wang et al., 2010).

# 2.2.2 Shared processes in the N and P cycle

The shared processes of N and P cycling include plant uptake, resorption, allocation, transfer from plant to the soil through litterfall, and transfer between organic litter and soil pools via biological mineralization and N, P biological immobilization (Fig. 2). Underlying these processes, TECO-CNP incorporates two key N-P interaction mechanisms: P uptake regulated by a nutrient balance scalar and a cost-benefit approach-based regulation of phosphatase production. To avoid duplication, the shared processes were described collectively.

The organic N ( $Q_N$ ) and P pools ( $Q_P$ ) are coupled with carbon pools through flexible stoichiometry within plant, litter, and soil pools. Inorganic nutrient components consist of one inorganic soil N pool ( $N_{min}$ ) and four inorganic soil P pools, including labile P ( $P_{lab}$ ), sorbed P ( $P_S$ ), secondary P ( $P_{SS}$ ), and occluded P ( $P_O$ ). Labile P represents readily bioavailable inorganic phosphate for biotic uptake and soil leaching. Sorbed P is weakly bound to soil surfaces in dynamic equilibrium with labile P. Through petrochemical processes, sorbed P transforms into secondary mineral P, which eventually becomes occluded P with minimal bioavailability. The key variables of N and P cycling are listed in Tables 2 and 3, respectively. Key parameter values were derived from site-specific field observations of plant functional traits and biogeochemical properties, as well as from validated studies chosen based on careful consideration of the ecosystem characteristics of the study site (Tables S1-S3, Table 4).

The initial size of the organic nutrient pool is determined by the size of the carbon pool and the carbon-to-nutrient ratio. The dynamics of organic nitrogen and phosphorus transfer from donor to recipient pools within plants, litter, and soil are coupled with carbon cycling through flexible stoichiometry. The dynamics of plant nutrient pools can be expressed as:

$$\frac{d}{dt}Q_{\chi,i}(t) = F_{new,\chi,i} - Q_{C,i} * \tau_i * R_{\chi,i}^{-1},$$
(17)

$$F_{new,\chi,i} = F_{new,C,i} * R_{\chi,i}^{-1} + (Q_{C,i} * R_{\chi,i,0}^{-1} - Q_{C,i} * R_{\chi,i}^{-1})$$
(18)

where subscript  $\chi = N$ , P,  $F_{new,\chi,i}$  represents the newly input nutrients from non-structural nutrient pool to sustain plant growth (Table 2),  $F_{new,C,i}$  is determined by the newly input carbon from the NSC pool to plant pool i and stoichiometric ratios (Eq. 25).  $R_{\chi,i,0}$  and  $R_{\chi,i}$  denote the initial and updated C:N (or C:P) ratios of plant pool i.  $Q_{C,i}$  and  $\tau_i$  represent the carbon pool size and turnover rate of plant pool i. The dynamically constrained nutrient redistribution process in plants (Eq. 18) follows the principles of stoichiometric homeostasis theory (Sterner & Elser, 2002) and helps avoid excessive flexibility in stoichiometry during model simulations (Meyerholt & Zaehle, 2015; Goll et al., 2017).

**Table 2.** Common variables for N and P cycle modeling.

| Variable                             | Description                                                    | Unit                              |
|--------------------------------------|----------------------------------------------------------------|-----------------------------------|
| F <sub>up,χ</sub>                    | Amount of nutrient uptake by plant roots                       | g m <sup>-2</sup> h <sup>-1</sup> |
| $F_{\text{res},\chi}$                | Amount of nutrient resorption before tissue litterfall         | $g m^{-2} h^{-1}$                 |
| $F_{\text{dm},\chi}$                 | Nutrient demand for plant growth                               | $g m^{-2} h^{-1}$                 |
| $F_{sp,\chi} \\$                     | Soil nutrient supply                                           | $g m^{-2} h^{-1}$                 |
| $uc_{root,\chi}$                     | Root uptake capacity                                           | $g m^{-2} h^{-1}$                 |
| $F_{new,\chi,i}$                     | Nutrient input for plant pool i                                | $g m^{-2} h^{-1}$                 |
| $F_{\chi,\text{min},x}$              | Mineralization fluxes of litter or soil pools, $x = m, j$      | $g m^{-2} h^{-1}$                 |
| $F_{\chi,imm,x}$                     | Immobilization fluxes of litter or soil pools, $x = m, j$      | $g m^{-2} h^{-1}$                 |
| $F_{\chi, \text{min}, \text{total}}$ | Total mineralization flux                                      | $g m^{-2} h^{-1}$                 |
| $F_{\chi,imm,total}$                 | Total immobilization flux                                      | $g m^{-2} h^{-1}$                 |
| $F_{\chi,net}$                       | Net mineralization flux                                        | $g m^{-2} h^{-1}$                 |
| $F_{\chi,in} \\$                     | Nutrient input to ecosystem                                    | $g m^{-2} h^{-1}$                 |
| $F_{\chi,loss}$                      | Nutrient loss from ecosystem                                   | $g m^{-2} h^{-1}$                 |
| $F_{\chi,leach}$                     | Nutrient loss through leaching                                 | $g m^{-2} h^{-1}$                 |
| $F_{\chi, \text{fert}}$              | Nutrient fertilization rate                                    | $g m^{-2} h^{-1}$                 |
| $F_{\chi,\text{dep}}$                | Nutrient atmospheric deposition rate                           | $g m^{-2} h^{-1}$                 |
| $F_{P2L,ij} \\$                      | Nutrient flux from plant pool $i$ to litter pool $j$           | $g m^{-2} h^{-1}$                 |
| $R_{\chi,i} \\$                      | Carbon: nutrient ratio of plant pool i                         | $g gC^{-1}$                       |
| $c_k$                                | Unit conversion factor for root uptake capacity                | unitless                          |
| $V_{runoff} \\$                      | Volume of drainage water                                       | $mm s^{-1}$                       |
| $\mathrm{D}_{\mathrm{soil}}$         | Soil depth                                                     | cm                                |
| $T_{soil} \\$                        | Soil temperature                                               | $^{\circ}\mathrm{C}$              |
| $f_{\chi,leach}$                     | Scalar for nutrient leaching                                   | unitless                          |
| Θ                                    | Volumetric soil water content                                  | $m^3 m^{-3}$                      |
| $L_{in,i}$                           | Tissue nutrient concertation stress factor of plant pool i     | unitless                          |
| $L_{sp}$                             | Nutrient uptake stress factor                                  | unitless                          |
| $L_{de}$                             | Nutrient limitation factor for decomposition                   | unitless                          |
| $L_{GP}$                             | Nutrient limitation scalar for plant growth                    | unitless                          |
| $f_{\chi,ratio}$                     | Nutrient concentration stress scalar affecting nutrient uptake | unitless                          |

<sup>310 \*</sup>  $\chi$  indicates N or P. Subscripts i, m, and j refer to the values in Table 1.

# **Table 3.** Specific variables in N and P cycle modeling.

| Variables          | Variables Description                                                                                                                        |                                |  |  |
|--------------------|----------------------------------------------------------------------------------------------------------------------------------------------|--------------------------------|--|--|
| N cycling specific |                                                                                                                                              |                                |  |  |
| $F_{N,fix}$        | N fixation rate                                                                                                                              |                                |  |  |
| $C_{\mathrm{fix}}$ | Carbon cost for biological N fixation                                                                                                        |                                |  |  |
| $F_{N,gas} \\$     | N loss in gaseous form                                                                                                                       |                                |  |  |
| P cycling specific |                                                                                                                                              |                                |  |  |
| K                  | Permeability of the soil to P                                                                                                                | m <sup>2</sup> h <sup>-1</sup> |  |  |
| $\alpha_{root}$    | Represents the fraction of the reduction in P concentration surrounding the roots relative to the initial concentration                      | unitless                       |  |  |
| $P_{lab}$          | Soil labile P                                                                                                                                | $gP m^{-2}$                    |  |  |
| P <sub>lab</sub> ' | Root surface soil labile P                                                                                                                   | $gP m^{-2}$                    |  |  |
| $\Delta P_{lab}$   | P concentrations in the soil solution at the root surface compared to the labile P in the surrounding soil outside the root's diffusive zone | gP m <sup>-2</sup>             |  |  |
| $P_S$              | Sorbed P                                                                                                                                     | $gP m^{-2}$                    |  |  |
| $P_{SS}$           | Secondary P                                                                                                                                  | $gP m^{-2}$                    |  |  |
| $P_{O}$            | Occluded P                                                                                                                                   | $gP m^{-2}$                    |  |  |
| $FP_{biomin} \\$   | P biochemical mineralization rate                                                                                                            | $gP m^{-2} h^{-1}$             |  |  |
| $FP_{diff} \\$     | Diffusion of P from the surroundings to the root surface                                                                                     | $gP m^{-2} h^{-1}$             |  |  |
| $F_{\mathrm{wea}}$ | P weathering rate                                                                                                                            | $gP m^{-2} h^{-1}$             |  |  |

**Table 4.** Parameters for nitrogen and phosphorus cycling in TECO-CNP.

| Short<br>name             | Value                 | Description                                                                                                                 | Reference  |
|---------------------------|-----------------------|-----------------------------------------------------------------------------------------------------------------------------|------------|
| N cycling                 |                       |                                                                                                                             |            |
| ken                       | 0.01                  | Empirical parameter for nitrogen concentration limitation (gN gC <sup>-1</sup> )                                            | Ref 1      |
| $\alpha N$                | 0.20                  | Fraction of N relocated before littering (Unitless)                                                                         | Ref 2      |
| F <sub>N,dep</sub>        | 3.60                  | N deposition (gN m <sup>-2</sup> yr <sup>-1</sup> )                                                                         | Ref 4      |
| $V_{\mathrm{fix}}$        | 1.67×10 <sup>-3</sup> | Maximum N fixation ratio (gN gC <sup>-1</sup> m <sup>-2</sup> h <sup>-1</sup> )                                             | Ref 5      |
| V <sub>max,N</sub>        | 5.40                  | Maximal root uptake capacity for N (μmol gC <sup>-1</sup> h <sup>-1</sup> )                                                 | Ref 6      |
| $k_{N,1} \\$              | 2.00×10 <sup>-3</sup> | Parameter to match the observed rate of increase in overall N uptake at high mineral N concentration ( $\mu$ mol $l^{-1}$ ) | Ref 6      |
| k <sub>N,2</sub>          | 98.00                 | For Michaelis-Menten constants, mineral N concentration at which uptake equals $\nu_{max}/2$ (µmol $1^{-1}$ )               | Ref 6      |
| P cycling                 |                       |                                                                                                                             |            |
| Vmax,P                    | 1.39                  | Maximal root uptake capacity for P (μmol gC <sup>-1</sup> h <sup>-1</sup> )                                                 | Ref 6      |
| $k_{P,1} \\$              | 0.01                  | Parameter to match the observed rate of increase in overall P uptake at high labile P concentration (µmol l <sup>-1</sup> ) | Ref 6      |
| $k_{P,2}$                 | 3.00                  | For Michaelis-Menten constants, labile P concentration at which uptake equals $\nu_{max}/2$ (µmol $l^{-1}$ )                | Ref 7      |
| $S_{\text{max}}$          | 133.00                | Maximum amount of sorbed P (gP m <sup>-2</sup> )                                                                            | Ref 8      |
| $K_s$                     | 64.00                 | An empirical parameter for describing the equilibrium between labile P and sorbed P (gP m <sup>-2</sup> )                   | Ref 8      |
| $\nu_{m}$                 | 2.05×10 <sup>-5</sup> | Rate constant of conversion from sorbed P to secondary P (gP m <sup>-2</sup> h <sup>-1</sup> )                              | Ref 1      |
| $\nu_{dis}$               | $2.40 \times 10^{-6}$ | Rate constant of conversion from secondary P to sorbed P (gP m <sup>-2</sup> h <sup>-1</sup> )                              | Calibrated |
| $\lambda_{\rm up}$        | 25.00                 | N cost of plant root P uptake (gN gP -1)                                                                                    | Ref 1      |
| $\lambda_{ptase}$         | 15.00                 | N cost of phosphatase production (gN gP <sup>-1</sup> )                                                                     | Ref 1      |
| $\kappa_{\text{m}}$       | 150.00                | Michaelis-Menten constant for biochemical P mineralization (gN gP <sup>-1</sup> )                                           | Ref 1      |
| $\nu_{\text{max}}$        | 0.02                  | Maximal specific rate of biochemical P mineralization (gP m <sup>-2</sup> h <sup>-1</sup> )                                 | Ref 1      |
| $k_{\text{cp}}$           | 0.0006                | Empirical parameter for phosphorus concentration limitation (gP gC <sup>-1</sup> )                                          | Ref 1      |
| $\alpha_P$                | 0.40                  | Fraction of P relocated before littering (Unitless)                                                                         | Ref 2      |
| $F_{wea} \\$              | 0.05                  | P weathering rate (gP m <sup>-2</sup> yr <sup>-1</sup> )                                                                    | Ref 1      |
| $F_{P,dep}$               | 0.06                  | Atmospheric P deposition rate (gP m <sup>-2</sup> yr <sup>-1</sup> )                                                        | Ref 4      |
| $r_{\rm d}$               | $3.10 \times 10^{5}$  | Root specific density (g biomass m <sup>-3</sup> )                                                                          | Ref 9      |
| $\mathbf{r}_{\mathrm{r}}$ | 2.90×10 <sup>-4</sup> | Fine root radius (mm)                                                                                                       | Ref 6      |
| $\mathbf{f}_1$            | 1.58                  | Empirical parameters for calculation of the tortuosity factor (Unitless)                                                    | Ref 10     |
| $f_2$                     | -0.17                 | Empirical parameters for calculation of the tortuosity factor (Unitless)                                                    | Ref 10     |
| $K_0$                     | 3.20×10 <sup>-6</sup> | Diffusion coefficient of phosphate in free water at 25 °C (m <sup>2</sup> h <sup>-1</sup> )                                 | Ref 11     |
| $\Theta_1$                | 0.12                  | relative water content (m <sup>3</sup> m <sup>-3</sup> )                                                                    | Ref 6      |
| $\alpha_P$                | 0.40                  | Fraction of P relocated before littering (Unitless)                                                                         | Ref 2      |

<sup>\*</sup> For reference codes, see Table S4.

Nutrients newly acquired from root uptake  $(F_{up,\chi})$  and tissue resorption  $(F_{res,x})$  enter the labile nutrient pool, which buffers the nutrient dynamics and mitigates imbalances between supply and demand (Weng et al., 2017). Thus, the dynamics of plant labile nutrient pools are modeled as:

$$\frac{d}{dt}NS_{\chi}(t) = F_{up,\chi} + F_{res,\chi} - \sum_{i} F_{new,\chi,i} . \tag{19}$$

Since the reproduction pool is designed as a long-term pool supporting a series of reproductive events, from flower bud formation to fruiting, no resorption is prescribed in this pool. The relocation of nutrients from senesced plant tissues  $(F_{res,\chi})$  is modeled as:

$$F_{res,\chi} = \sum_{i} \alpha_{\chi} \times Q_{c,i} * \tau_{i} * R_{\chi,i}^{-1} \text{ (i \neq reproduction)},$$
 (20)

- where  $\alpha_{\chi}$  is the resorption rate and the second term represents the loss of carbon from plant pool *i* (Table 4). We assume that the different plant organs have the same and fixed resorption rate to simplify this process. Additionally, we prescribe a higher resorption rate for P at 0.4 compared to N at 0.2, considering the higher phosphorus use efficiency in the P-limit habitat (Xu et al., 2020).
- Litter nutrient dynamics is given by:

$$\frac{d}{dt}Q_{\chi,j}(t) = F_{P2L,ij} - Q_{C,j} * \tau_j * R_{\chi,j}^{-1}, \qquad (21)$$

where  $F_{P2L,ij}$  represent the nutrient flux from plant pool i to metabolic litter (j = 5) and structure litter (j = 6):

$$F_{P2L,ij} = \begin{cases} (1 - \alpha_{\chi})Q_{C,i} * R_{\chi i}^{-1} * \tau_{i} * r_{i,j}, i = 1, 2, 3\\ Q_{C,i} * R_{\chi i}^{-1} * \tau_{i} * r_{i,j}, i = 4 \end{cases},$$
(22)

- where  $r_{i,j}$  represents the fraction of plant carbon to different litter pools.
- The TECO-CNP model exclusively considers the active uptake of inorganic P through specialized transporters on the root surface (Schachtman et al., 1998), as inorganic P is the form most readily absorbed by plants (Bieleski, 1973). Plants possess specific transporters and mechanisms dedicated to transmembrane transport, ensuring they can acquire P even from soil solutions with low P concentrations, where the P concentration can be as low as

one-thousandth of the intracellular concentration (Schachtman et al., 1998). Therefore, we assume that plants absorb only inorganic P from the soil. Similarly, we also only consider the plant uptake of inorganic N. Soil labile nutrients taken up by plants are generally contingent upon both nutrient demand for growth (Wang et al., 2010) and root uptake capacity (Grant et al., 1999, 2001; Goll et al., 2017) that are related to root morphology and soil nutrient concentrations. The nutrient demand-supply scheme has been widely employed in most coupled C-nutrient models (Achat et al., 2016). We assume plants will not consume nutrients beyond their luxury consumption demand for assimilating nutrients (Van Wijk et al., 2003; Chapin, 1980). Therefore, the  $F_{up,x}$  is determined by either the nutrient demand ( $F_{dm,\chi}$ ) or the nutrients supplied by soil ( $F_{sp,\chi}$ ), whichever is lower:

$$F_{up,\chi} = \begin{cases} F_{dm,\chi} & \left( F_{dm,\chi} < UC_{root,\chi} \right) \\ F_{sp,\chi} & \left( F_{dm,\chi} > UC_{root,\chi} \right) \end{cases}$$
(23)

The  $F_{dm,\chi}$  is determined by the invested carbon for newly formatted tissues  $(NPP_i)$  and C:nutrient ratios. The actual demand is considered as the difference between the demand for growth and resorption capacity:

$$F_{dm,\chi} = \sum_{i} \frac{F_{new,C,i}}{R_{\chi,i}} - F_{res,\chi} , \qquad (24)$$

$$F_{new,C,i} = NPP_a * b'_{C,i}, (25)$$

where  $NPP_a$  represents the net primary productivity derived from actual plant growth (Eq. 1),  $b'_{C,i}$  denotes the  $b_{C,i}$  (Eqs. 9-11) specifically influenced by the leaf phenology (Weng & Luo, 2008).

The nutrients supplied to plants from the soil depend not only on the amount of P in the soil but also on soil conditions and the root uptake capacity. We implemented the function of  $F_{sp,\chi}$  as described by Goll et al. (2017), and it is calculated by the function of root biomass  $(BM_{root})$ , and root uptake capacity  $(uc_{root,\chi})$ , soil temperature scalar  $(f_T)$  and the nutrient balance scalar  $(f_{\chi,ratio})$  as follows:

$$F_{sp,\chi} = BM_{root} * uc_{root,\chi} * f_T * f_{\chi,ratio}.$$
 (26)

The linear index scalar  $f_{\chi,ratio}$  regulates the balance between C, N, and P by constraining nutrient uptake rates based on prescribed maximum ratios (Eqs. 27-28), thereby preventing resource overconsumption (Goll et al., 2017). Experiments have shown that N addition enhances the uptake of both N and P, suggesting a benefit for P uptake when more N is available (Zhu et al., 2021). Thus, we assume that the dependence of P uptake on the plant P:N ratio is modeled as a function of the P:N ratio of both the plant and its leaves, thereby capturing the essential N-P interaction through stoichiometric regulation. This regulatory mechanism helps prevent excessive P uptake, which would constitute luxury consumption for the plant (Schachtman et al., 1998). Similarly, if nitrogen uptake exceeds the plant's requirements, it also constitutes luxury consumption. Therefore, to avoid luxury absorption and nutrient accumulation, the uptake of N (or P) by roots needs to be regulated based on the N:C (or P:N) ratios within plant tissues (Goll et al., 2017). The maximum uptake occurs when the leaf N:C (or P:N) ratio is equal to the minimum leaf N:C (or P:N) ratio, which is calculated using a minimum function:

$$f_{P,ratio} = \min\left(\max\left(\frac{pn_{plant} - pn_{leaf,max}}{pn_{leaf,min} - pn_{leaf,max}}, 0.0\right), 1.0\right), \tag{27}$$

$$f_{N,ratio} = \min\left(\max\left(\frac{nc_{plant} - nc_{leaf,max}}{nc_{leaf,min} - nc_{leaf,max}}, 0.0\right), 1.0\right), \tag{28}$$

where  $pn_{leaf,max}$  and  $pn_{leaf,min}$  are prescribed maximum and minimum values of leaf P:N ratios,  $nc_{leaf,max}$  and  $nc_{leaf,min}$  are prescribed maximum and minimum values of leaf N:C ratios.

The root nutrient-uptake capacity function  $(uc_{root,\chi})$  incorporates both linear and Michaelis-Menten components to accurately represent the uptake process, considering the low-affinity and high-affinity transporter systems operating in parallel for a given solute concentration (Goll et al., 2017). Notably, the root uptake capacity for soil labile P  $(u_{root,P})$  considers the replenishment of P from soil around the roots to root surfaces (Goll et al., 2017) rather than the total labile P in soil volume (Schachtman et al., 1998; Johnson et al., 2003). Hence, the calculation of root uptake capacity for N and P can be expressed as follows:

$$u_{root,P} = v_{max,P} * P'_{lab} \left( \frac{k_{P_{m1}}}{c_k} + \frac{1}{P_{lab} + c_k k_{P_{m2}}} \right), \tag{29}$$

$$u_{root,N} = v_{max,N} * N_{min} \left( \frac{k_{N_{m1}}}{c_k} + \frac{1}{N_{min} + c_k k_{N_{m2}}} \right),$$
 (30)

where  $v_{max,\chi}$  is the maximum uptake capacity (Table 4).  $N_{min}$  and  $P_{lab}$  is the soil mineral N pool and labile P pool.  $P'_{lab}$  represents the dissolved labile P concentration at the root surface and depends on the diffusion of soil labile P from the soil surrounding the roots to the root surface (Table 3; Eq. 54).  $c_k$  is a unit conversion factor using the soil-type specific parameter for soil moisture content at saturation as an approximation of pore space following Smith et al. (2014).  $k_{\chi_{m1}}$  was chosen to match the observed rate of increase in overall P uptake at high dissolved labile P concentration (low-affinity transporter), and  $k_{x_{m2}}$  is a parameter for Michaelis-Menten constants, dissolved phosphorus concentration at which uptake equals  $\frac{v_{max}}{2}$ .

Mineralization and immobilization processes occur concurrently. Nutrient mineralization fluxes are estimated from the decomposition of litter and soil organic matter, assuming that C, N, and P mineralize at similar rates (Wang et al., 2010; Yang et al., 2014). The mineralization rate is determined by multiplying the litter and soil C pool turnover fluxes by the nutrient-to-carbon ratio. This can be mathematically represented by the following equations:

$$F_{\chi,min,j} = Q_C(t)\tau_j \xi(t) R_{\chi,j}^{-1} , \qquad (31)$$

$$F_{\chi,min,m} = Q_C(t)\tau_m \xi(t) R_{\chi,m}^{-1},$$
 (32)

where  $Q_{\mathcal{C}}(t)\tau\xi(t)$  estimates the carbon decomposition rate under environmental stress for 398 399 litter or soil pool. The total mineralization  $(F_{\gamma,min,total})$  is estimated as the sum of 400 mineralization rate for each pool, which can be expressed as follows:

$$F_{\chi,min,total} = \sum_{j} F_{\chi,min,j} + \sum_{m} F_{\chi,min,m} . \tag{33}$$

Nutrients are immobilized during the decomposition process of litter and SOM, 402 ultimately entering the SOM pools. Consequently, only three SOM pools can be the receiving pools. The dependency of immobilization rates on the ratios of the receiving pools, under the assumption of approximately constant stoichiometry ratios of SOM pools (Tian et al., 2010; McGroddy et al., 2004), is described as:

$$F_{\chi,imm,m} = \sum_{jm} f_{L2S,jm} \xi(t) \tau_j X_j(t) R_{\chi,m}^{-1} + \sum_{mm} f_{S2S,mm} \xi(t) \tau_m X_m(t) R_{\chi,m}^{-1} , \qquad (34)$$

- where  $R_{\chi,m}^{-1}$  represent the N:C ratio ( $\chi = N$ ) or the P:C ratio ( $\chi = P$ ) of the existing SOM.
- The total amount of immobilization is then calculated as follows:

$$F_{\chi,imm,total} = \sum_{m} F_{\chi,imm,m} . \tag{35}$$

Therefore, the net nutrient mineralization is calculated by the difference of total mineralization and total immobilization:

$$F_{\chi,net} = F_{\chi,min,total} - F_{\chi,imm,total} . \tag{36}$$

When net mineralization is negative, the decomposition rate is limited by nutrient availability,  $L_{NP}$ . Since the N:C ratio of the soil pool is higher than that of the litter pool, microbes extract inorganic nitrogen from the soil mineral N pool, leading to negative net mineralization and a  $L_{NP}$ . value less than one. A similar approach has been applied in the CASA-CNP model (Wang et al., 2007).

Plants and microorganisms utilize dissolved inorganic N and P from the soil to fulfill their growth requirements (Vitousek et al., 2010). We assume microbial processes modulate nutrient availability for plants (Jiang et al., 2024b; Pellitier et al., 2023; Jonasson et al., 1999), i.e., the nutrient limitation on plant growth will be alleviated if the net mineralization is positive. Furthermore, the competition between plants and microorganisms for nutrients can be simplified by emphasizing the sequence of immobilization and plant uptake (Achat et al., 2016). In the TECO-CNP model, immobilization takes precedence in nutrient access through the decomposition of litter and soil organic matter. A similar method was used in many models, e.g., models of the CENTURY family (e.g., Parton et al., 1988); O-CN (Zaehle and Friend 2010); ORCHIDEE (revision 4520; Goll et al., 2017). This also aligns with recent findings regarding the competition between plants and microbes under elevated CO<sub>2</sub>. (Keane et al., 2023). Specifically, in the acidic grassland, aboveground productivity and P uptake declined by

428 11% and 20%, respectively, while P immobilization into microbial biomass increased by

429 36%.

### 431 2.2.3 Distinct processes in N and P cycle

The dynamic of soil inorganic N  $(N_{min})$  is described as:

$$\frac{dt}{d}N_{min} = F_{fix} + F_{N,fert} + F_{N,dep} - F_{N,leach} - F_{N,gas},$$
(37)

where  $F_{fix}$ ,  $F_{N,fert}$  and  $F_{N,dep}$  represent the biological N<sub>2</sub>-fixation, atmospheric N

deposition, and biological N fixation (Tables 2, 3).  $F_{N,leach}$  and  $F_{N,gas}$  represent the N

leaching and gaseous N loss.

Biological nitrogen fixation, a dominant source of new nitrogen in terrestrial

ecosystems (Chapin et al., 2011; Vitousek et al., 2013), is performed by N<sub>2</sub>-fixing

symbionts in plant roots (i.e., symbiotic N<sub>2</sub>-fixation; Vitousek et al., 2002; Augusto et al.,

2013). This process enhances nitrogen availability when carbon is sufficient for additional

nutrient acquisition (Fisher et al., 2010), which is given by:

$$F_{fix} = v_{fix} * f_{nsc} * NSC * f_N,$$
(38)

where  $v_{fix} = 0.00167$  (gN gC<sup>-1</sup> m<sup>-2</sup> h<sup>-1</sup>) is the maximum N fixation rate.  $v_{fix}$  is chosen

based on estimates ranging from 58 Tg N yr<sup>-1</sup> (Vitousek et al., 2013) to 100 Tg N yr<sup>-1</sup>

(Wiltshire et al., 2021) for a global NPP of 60 Pg C yr<sup>-1</sup>. The term  $f_{nsc} * NSC$  represents

the limitation of NSC on nitrogen fixation, implicitly capturing the carbon constraint on

this process (Chou et al., 2018; Taylor et al., 2021). To prevent unrealistic nitrogen fixation,

a scaling function  $(f_N)$  is applied, as nitrogen fixation is an energy-intensive process

(Gutschick, 1981; Goll et al., 2017). The  $f_N$  is calculated as:

$$f_{N} = \begin{cases} \frac{N_{max} - N_{min}}{N_{max}} & N_{max} < N_{min} \\ 0 & otherwise \end{cases}$$
 (39)

The carbon cost for biological N fixation is calculated by a function of soil temperature

$(T_{soil})$  with the observed carbon cost range (Fisher et al., 2010):

$$C_{fix} = -6.25 * \left( \exp\left( -3.62 + 0.27 * T_{soil} * \left( 1 - 0.5 \left( \frac{T_{soil}}{25.15} \right) \right) \right) - 2 \right). \tag{40}$$

- Nitrogen loss occurs in two pathways: gaseous loss  $(FN_{gas})$ , and leaching  $(F_{N,leach})$ .
- Losses from denitrification and volatilization are not distinguished separately. Both are
- proportional to the availability of soil mineral N  $(N_{min})$ . The expression of N leaching is:

$$F_{N,leach} = f_{N,leach} \frac{V_{runoff}}{D_{soil}} N_{min} , \qquad (41)$$

- where  $f_{N,loss} = 0.001$  and  $f_{N,leach} = 0.5$ ,  $V_{runoff}$  is the soil surface runoff and  $D_{soil}$  is the
- soil depth. Moreover, the gaseous loss is dependent on the soil temperature and soil mineral
- N. The equation is:

$$F_{gas} = f_{N,loss} e^{\frac{(T_{soil} - 25)}{10}} N_{min}. (42)$$

The specific processes of the P cycle include biochemical mineralization, weathering, the dynamics of different inorganic soil P components, and the diffusion pathways of soil labile P. In addition to biological mineralization, organic P can be mineralized through direct cleavage by extracellular enzymes produced by plant roots and other organisms (McGill and Cole, 1981). This process decouples the P cycle from the C and N cycles, serving as an adaptive mechanism that can be enhanced under P-limited conditions (Lambers et al., 2006). This decoupling allows for phosphorus acquisition from organic matter without releasing carbon dioxide. We consider this process an N-consuming one, aiming to represent the chemical characteristic that phosphatases are N-rich enzymes and their production in plants can be N-limited (Treseder and Vitousek, 2001; Wassen et al., 2013). The biochemical mineralization of P can be expressed by:

$$FP_{biomin,m} = \frac{\nu_{max}(\lambda_{up} - \lambda_{ptase})}{\lambda_{up} - \lambda_{ptase} + \kappa_m} \sum_{m} K_m Q_{P,m}, \qquad (43)$$

where  $v_{max}$  is maximal specific rate of biochemical P mineralization.  $\lambda_{up}$  is N cost of plant root P uptake.  $\lambda_{ptase}$  is the N cost of phosphatase production,  $\kappa_m$  is the Michaelis-Menten constant for biochemical P mineralization.  $K_m$  and  $Q_{P,m}$  represent turnover rate and phosphorus pool size of slow (m = 8) and passive pools (m = 9). Phosphatase production

is activated when  $\lambda_{up} > \lambda_{ptase}$ , reflecting N regulation of P acquisition strategy by plants, 472 similar to the cost-benefit approach established in existing coupled carbon-nutrient models 473 (Wang et al., 2007; Houlton et al., 2008). This modeling approach aligns with findings that 474 nitrogen addition significantly increases phosphatase activity (Schleuss et al., 2020; 475 Marklein et al., 2012), potentially through enhanced phosphorus limitation and elevated 476 plant nitrogen status, which favor investment in the phosphatase enzyme. While direct field 477 quantification of biochemical mineralization rates is not yet possible, this mechanistic 478 representation becomes particularly important for predicting ecosystem responses to 479 elevated CO2 and enhanced N deposition, where enhanced biochemical mineralization of 480 soil organic P may facilitate additional plant growth (Jiang et al., 2024a).

The soil P loss from soil organic P pools can be simulated by the following equations:

$$F_{P,out,m}(t) = Q_{C,m} * \tau_m * R_{\chi,m}^{-1} + FP_{biomin,m}.$$
 (44)

The term  $FP_{biomin,m}$  equals 0 when m = 7 as organic P losses through biochemical mineralization only occur in two soil pools with slow turnover rates (slow and passive pools; Wang et al., 2010).

The external phosphorus input  $(F_{P,in})$  is modeled as:

$$F_{Pin} = F_{weg} + F_{Pfert} + F_{Pden}$$
, (45)

where  $F_{wea}$ ,  $F_{P,fert}$ , and  $F_{P,dep}$  represent phosphorus input rates from weathering, fertilization, and deposition. Based on the soil texture at the Tiantong site (Song & Wang, 1995), the weathering rate is set to 0.005 (gP m<sup>-2</sup> year<sup>-1</sup>) (Wang et al., 2010). The deposition rate of phosphorus has been set to 0.06 (gP m<sup>-2</sup> year<sup>-1</sup>) (Zhu et al., 2016).

Labile phosphorus ( $P_{lab}$ ) can be directly utilized by plants or microorganisms and adsorbed onto soil particles, organic matter, and other minerals as adsorbed phosphorus ( $P_S$ ) (Vitousek et al., 2010). The assumption is made that the rapid equilibration of  $P_{lab}$  with  $P_S$  occurs within a timestep of less than one hour (Olander and Vitousek, 2005). For the 1-hour time step used in our study, we therefore assume that  $P_{lab}$  and  $P_S$  are in a state of equilibrium. The equilibrium assumption is applied extensively (e.g., Wang et al., 2007; Yang et al., 2014). The relationship between them is described by a Langmuir isotherm (Barrow, 2008):

$$P_S = \frac{S_{max}P_{lab}}{K_c + P_{lab}},\tag{46}$$

where  $S_{max}$  is the maximum amount of sorbed P in the soil, and  $K_s$  is the empirical constant representing the tendency of soil labile P to be sorbed.  $S_{max}$  and  $K_s$  is set as 133 and 64 (Wang et al., 2010), respectively, according to the soil sorption capacity and substrate age (Olander and Vitousek, 2005) at the Tiantong site. The differential form of Eq. 46 is:

$$\frac{dP_S}{dt} = \frac{S_{max}P_{lab}}{(K_S + P_{lab})^2} \frac{dP_{lab}}{dt} \,. \tag{47}$$

Assuming equilibrium between  $P_{lab}$  and  $P_S$ , we can model the simultaneous changes in  $P_{lab}$  and  $P_S$  as follows:

$$\frac{d(P_S + P_{lab})}{dt} = F_{P,net} + F_{P,in} + F_{P,biomin} - F_{up,P} - F_{P,leach} - \nu_m P_S , \qquad (48)$$

$$F_{P,net} = F_{P,min,total} - F_{P,imm,total} , (49)$$

where  $F_{P,net}$  is the net mineralization of litter and soil phosphorus pool,  $F_{P,biomin}$  is the P flux from biochemical mineralization,  $U_P$  represents the plant uptake of P,  $F_{P,leach}$  represents the loss of labile P from leaching (Eq. 53), and  $v_m$  is the rate constant for the transformation of sorbed P to secondary P. Based on Eq. 48 and Eq. 49, the dynamics of labile phosphorus can be expressed as follows:

$$\frac{dP_{lab}}{dt} = \left(F_{P,net} + F_{P,in} + FP_{biomin} - F_{up,P} - F_{P,loss} - \nu_m P_S\right) \frac{1}{1 + \frac{S_{max}P_{lab}}{(K_S + P_{lab})^2}},\tag{50}$$

The use of solution P would be theoretically more appropriate, as previous studies have shown that models operating at very fine temporal resolutions (hourly or finer) may require distinction between labile and solution phosphorus pools (Reed et al., 2015; Yang et al., 2013). However, implementing this simulation approach is currently not feasible due to limited data availability. Some synthesis studies (Yang et al., 2013; Hou et al., 2018) have indicated that most experimental measurements report total labile P, without separating it into distinct fractions. Additionally, previous studies have demonstrated strong correlations between these fractions. For example, strip- and NaHCO<sub>3</sub>-extracted

inorganic P are positively correlated and exhibit similar temporal patterns during experimental periods (Hou et al., 2019). Due to these reasons, we adopted labile P as our primary plant-available phosphorus pool in our model.

Secondary mineral phosphorus ( $P_{SS}$ ) can be dissolved and enter the labile P pool or encapsulated by iron oxides to form closed-P ( $P_o$ ; Walker & Syers, 1976; Vitousek et al., 2010). The dynamics of  $P_{SS}$  and  $P_o$  can be modeled as:

$$\frac{dP_{SS}}{dt} = \nu_m P_S - \nu_{dis} P_{SS} - \nu_o P_{SS} , \qquad (51)$$

$$\frac{dP_o}{dt} = \nu_o P_{SS} - \nu_{re} P_o , \qquad (52)$$

where  $v_{dis}$  and  $v_o$  is the rate constant for the conversion of secondary P to labile and sorbed P, and occluded P, respectively.  $v_{re}$  is the rate constant for occluded P re-entering the cycle as bioavailable phosphorus, indicating that occluded phosphorus can transition back into available forms (Huang et al., 2014; Schubert et al., 2020). In this study, we assume that the formation of occluded P pool and loss of occluded P can be considered negligible within the short timescale of simulations (Weihrauch & Opp, 2018). P leaching from the soil inorganic labile pool is proportional to the availability of soil labile P. Description of P leaching below:

$$F_{P,leach} = f_{p,leach} \frac{V_{runoff}}{D_{soil}} P_{lab} , \qquad (53)$$

where  $V_{runoff}$  is the value of runoff,  $D_{soil}$  is the soil depth.  $f_{p,leach}$  is an empirical parameter for P leaching, representing the fraction of soil mineral P for leaching.

Notably, due to the low mobility of phosphorus in the soil (Vitousek et al., 2010), the actual P concentration that roots can absorb depends on the diffusion of P from the surrounding soil to the root surface ( $P'_{lab}$ ). This finding is consistent with the experimental evidence that roots primarily acquire most inorganic phosphorus through diffusion along concentration gradients (Laliberté et al., 2015). Thus, the root uptake capacity for soil labile P ( $u_{root,P}$ ) considers the replenishment of P from soil around the roots to root surfaces (Schachtman et al., 1998) rather than the total labile P in soil volume (Johnson et al., 2003).

Thus, the root surface P concentration is calculated by the following equation:

$$P'_{lab} = a_{root} * \frac{P_{lab}}{\Theta}, \tag{54}$$

- where  $\theta$  is the volumetric soil water content and  $a_{root}$  representing the fraction of the
- reduction in P concentration surrounding the roots relative to the initial concentration.
- $a_{root}$  is updated after plant uptake as:

$$\frac{\mathrm{d}a_{root}}{\mathrm{d}t} = \frac{FP_{diff} - F_{up,P}}{P_{lab}},\tag{55}$$

- where  $FP_{diff}$  is the diffusion of P from the surroundings to the root surface, which is the
- function of the permeability of the soil to P (K) and the difference in the P concentrations
- between the soil solution at the root surface and the labile P in the surrounding soil volume
- outside the diffusive zone around the root ( $\Delta P_{lab}$ )

$$FP_{diff} = -K * \Delta P_{lab} . ag{56}$$

$\Delta P_{lab}$  can be described as:

$$\Delta P_{lab} = (a_{root} - 1) \frac{P_{lab}}{\theta}. \tag{57}$$

- The K has been calculated analogously to the diffusion coefficient of phosphorus in soils
- following Barraclough and Tinker (1981), which accounts for the increased path length in
- soil using a tortuosity factor  $(f_t)$ , and it is a broken-line function of the volumetric soil
- water content ( $\Theta$ ). The K and  $f_t$  can be calculated based on the following equations:

$$K = K_0 c_\theta \theta t f \frac{1}{r_{diff}}, \tag{58}$$

$$f_{t} = \begin{cases} f_{1}\theta + f_{2} & for \, \theta \geq \theta_{1} \\ \frac{\theta(f_{1}\theta + f_{2})}{\theta_{1}} & otherwise \end{cases}, \tag{59}$$

- where  $\theta_1$  is soil water content at which the two functions intersect according to Barraclough
- and Tinker (1981),  $f_1$  and  $f_2$  are empirical parameters (Barraclough and Tinker, 1981),  $D_0$
- is diffusion coefficient in free water,  $c_{\theta}$  is a unit conversion factor,  $r_{diff}$  is diffusion path,
- which can be calculated from the function of root length density (*RLD*, Bonan et al., 2014):

$$r_{diff} = \min(0.1, (\pi R L D)^{0.5})$$
 (60)

We assume that the diffusion path can be approximated as half the average distance between roots. We limit the diffusion path length to 0.1 m because the influence of active P uptake by roots on soil P concentrations is negligible beyond a distance of 10 cm (Li et al., 1991). *RLD* is given by:

$$RLD = \frac{B_{root}^*}{r_d \pi r_r^2},\tag{61}$$

where  $r_d$  is the root-specific density and  $\pi r_r^2$  is the cross-sectional area calculated from the fine root radius,  $r_r$ , and  $B_{root}^*$  is the root biomass density per unit soil volume.

The tension between high carbon sink capacity and nutrient limitations in subtropical

#### 2.3 Model validation

## 2.3.1 Study site

forests warrants detailed investigation to understand the role of nutrients in carbon cycling processes in these regions. To this end, we selected a mature subtropical evergreen broadleaf forest in eastern China, located at the Zhejiang Tiantong Forest Ecosystem National Observation and Research Station (Tiantong, 29°48' N, 121°47' E, Fig. 3) for the newly developed model. The Tiantong forest has been preserved free from human disturbance since the mid-twentieth century. The average annual temperature in Tiantong is 17°C, and the annual precipitation is 1600 mm (Cui et al., 2022). The soil type is mainly mountainous yellow-red soil, with the parent material primarily composed of Mesozoic sedimentary rocks, acidic igneous rocks, and residual weathering products of granite (Song & Wang, 1995). Research conducted at this site revealed the dominant role of soil phosphorus in driving variations in plant functional traits (Cui et al., 2022), indicating phosphorus deficiency in this mature forest. Consequently, this phosphorus-limited mature subtropical forest, with abundant field observations, can contribute to the development of carbonnutrient coupling models and further explore phosphorus-limited carbon cycling processes within the ecosystem through the integration of modeling and experiments.

**Figure 3. Schematic diagram of the observation system at Tiantong subtropical evergreen forest (29°5' N, 121°5' E).** The system comprises: (1) a forest dynamic plot for monitoring ecosystem state variables, including stoichiometric ratios, plant traits, and C, N, P pools and fluxes. These measurements were conducted in a 5-ha subplot of the whole plot. The asterisk (\*) indicates manually measured periodic fluxes. And (2) an eddy covariance (EC) flux tower providing half-hourly NEE measurements, from which GPP and ER were derived. These observations were used for parameterizing and evaluating the TECO-CNP model. Detailed measurement protocols are described in the Methods section, and specific variable applications are listed in Tables S1-S3. The 3D visualization of the study site was created in Blender (v4.2.1) using topographic data sourced from OpenTopography (<a href="https://opentopography.org">https://opentopography.org</a>).

# 2.3.2 Data collection and site parameterization

The data used for model calibration and validation were primarily derived from our field measurements and literature focusing on the same site (Fig. 3). The forcing data for TECO are collected at 1-hour intervals from field-based meteorological observations at the study site, include precipitation (mm), relative humidity (%), air and soil temperatures (°C),

vapor pressure deficit (Pa), wind speed (m s<sup>-1</sup>), and photosynthetically active radiation (µmol m<sup>-2</sup> s<sup>-1</sup>). Forcing data from 2001 was used for model spin-up.

Site-specific parameters that can be empirically measured are derived from field observations at the study site, including both our measurements and values reported in previous studies. Plant traits, including specific leaf area (SLA, cm² g⁻¹), leaf area index (LAI, m² m⁻²), plant height (H, m), maximum rate of carboxylation (Vcmax, µmol m⁻² s⁻¹), maximum rate of electron transport (Jmax, µmol m⁻² s⁻¹) and leaf P concentration (Leaf P, g m⁻²), were measured at the species level in the forest dynamic plot, and scale up to community-level traits using the community-weight mean method (for detailed sampling methods, refer to Cui et al., 2022). Plant stoichiometry ratios were derived from area-based C, N, and P concentrations from both our measurements and previous studies at Tiantong (Zhou et al., 2020). N and P resorption efficiencies were determined based on dominant species (i.e., *Schima superba*, *Lithocarpus glaber*) at the Tiantong site (Xu et al., 2020). The observed data used for model parameterization are presented in Tables S1-S3.

Parameters not readily accessible through field measurements are estimated using standard procedures that have been extensively validated in other modeling studies, with appropriate selection based on the characteristics of Tiantong. For example, Tiantong forest soils are classified as *Ultisols* (Song & Wang, 1995), which directly informed our selection of phosphorus weathering rates and inorganic P dynamics parameters (e.g., K<sub>s</sub> and S<sub>max</sub>, Table 4). Similarly, the subtropical evergreen broadleaf forest vegetation type guided our parameterization of phosphorus mineralization (Wang et al., 2010) and allocation processes (Arora & Boer, 2005). External inputs of N and P, including deposition and weathering, were assumed to occur at constant rates. Deposition rates for N and P were prescribed based on the observed range (Zhu et al., 2016). Specific parameterization settings are described in Table 4, along with the accompanying process descriptions.

The observed pool sizes and fluxes primarily serve as a basis for model evaluation and as references for model initialization. Soil inorganic pools of mineral N and labile P were determined from 0-10 cm soil samples collected in 2023 from a nearby forest stand of similar age (~200 yr) dominated by the same species (*Schima superba* and *Castanopsis fargesii*) as the Tiantong forest dynamic plot. Labile P is the soil inorganic phosphorus

fraction that can be extracted by resin and NaHCO<sub>3</sub>. Sampling employed a five-point design with three replicates per point. Additionally, secondary P and occluded P refer to the measured moderately labile inorganic phosphorus (extracted by NaOH) and moderately stable inorganic phosphorus (extracted by HCl), respectively, at the Tiantong site (Wang, 2022).

Soil C, N, and P were measured using systematic sampling across 185 grid points (each 20 × 20 m) within the permanent Tiantong forest dynamic plot (Fig. 3). At each grid point, soil samples were collected at three depth intervals (0-20, 20-40, 40-60 cm) in 2017 using a 5 cm diameter auger, with three replicates per depth. Additionally, observed plant pools and fluxes, including fluxes from plant to litter and soil respiration, used for model evaluation and their sources are listed in Tables S1-S3. Quality-controlled hourly eddy covariance measurements of gross primary productivity (GPP), ecosystem respiration (ER), and net ecosystem exchange (NEE) were obtained from the on-site flux tower for the year 2021.

All model configurations used identical site-specific parameter sets obtained according to the methods described above. Although a previous study has highlighted the necessity for model-specific reparameterization (Wang et al., 2022), we adopted a consistent parameterization approach across all configurations. This follows common practice in land surface model development studies, where uniform parameterization is essential for isolating the effects of different nutrient coupling schemes.

#### 2.3.3 Data assimilation

We specifically optimized carbon-related parameters for the CNP configuration only, utilizing GPP, ER, and NEE data from 2021 at the study site, to evaluate the effectiveness of the CNP structure coupled with a data assimilation algorithm. Based on the initial carbon pool sizes from the spin-up process, a preliminary sensitivity analysis was first conducted to support the selection of target parameters for data assimilation. We focused on parameters that determine carbon input and retention (Table 6), including SLA,  $V_{cmax}$ , and temperature sensitivity ( $Q_{10}$ ), which showed high sensitivity in the analysis (Table S6). Additionally, our parameter selection strategy included all carbon pool turnover parameters ( $T_1$ - $T_9$ ), as these govern carbon residence times and are crucial for matching observed pool

dynamics, regardless of their sensitivity indices. The prior range of parameters was prescribed according to the situ measurement or assumed as the range of the distribution to be  $[\theta_0/3, 3\theta_0]$ , where  $\theta_0$  is the default value. Using the Bayesian probabilistic inversion approach, we estimated the posterior distribution of model parameters based on prior knowledge of the parameters.

Bayesian probabilistic inversion approach is based on Bayes' theorem:

$$p(\theta|Z) \propto \frac{p(Z|\theta) \times p(\theta)}{p(Z)},$$
 (61)

where  $p(\theta|Z)$  is the posterior distribution of the parameters  $\theta$  given the observations Z. Here, we assume that the prior knowledge of parameter distribution  $p(\theta)$  is uniformly distributed.  $p(Z|\theta)$  is the likelihood function for a parameter set calculated with the assumption that each parameter is independent from all other parameters and has a normal distribution with a zero mean:

$$p(Z|\theta) \propto \exp\left\{-\sum_{t \in Z_i} \frac{[Z_i(t) - X(t)]^2}{2\sigma^2(t)}\right\},\tag{62}$$

where  $Z_i(t)$  is the observations of carbon fluxes at time t, X(t) is the simulated corresponding variable, and  $\sigma(t)$  is the standard deviation of the observation set.

Posterior probability distributions of the parameters were obtained using a Metropolis-Hastings (M-H) algorithm within the Markov Chain Monte Carlo (MCMC) framework. The posterior parameter distribution represents our updated knowledge about parameter values after incorporating observational data through Bayesian inference, quantifying both the most likely parameter estimates and their associated uncertainties. The detailed description of the M-H algorithm can be found in Xu et al. (2006). In brief, the M-H algorithm consists of iterations that alternate between a proposing step and a moving step. In the proposing step, a new parameter set  $\theta^{new}$  is proposed based on the previously accepted parameter set  $\theta^{old}$  and a proposal distribution  $(r \times (\theta_{max} - \theta_{min})/D)$ :

$$\theta^{new} = \theta^{old} + r \times (\theta_{max} - \theta_{min})/D$$
, (63)

where  $\theta_{max}$  and  $\theta_{min}$  corresponding to the upper and lower values of prescribed ranges, r is a random variable between -0.5 and 0.5, and D is used to control the proposed step size

and was set to 5 (Xu et al., 2006). The new set of parameter values would be accepted when  $\frac{p(\theta^{new}|Z)}{p(\theta^{old}|Z)}$  is equal or greater than a uniform random number from 0 to 1 (Xu et al., 2006).

We get 10,000 accepted samples from the MCMC chain. The first 5000 accepted samples were discarded, considering the burn-in period. We randomly selected 1,000 parameter sets from the accepted space to run the simulations in 2021. The mean and maximum likelihood estimations are calculated to compare the parameters.

# 2.3.4 Model performance evaluation

The state variables estimations from three nutrient coupling configurations of TECO-CNP: (1) carbon-only (C-only), (2) carbon-nitrogen coupled (CN), and (3) carbon-nitrogenphosphorus coupled (CNP) are evaluated against observations. Model initialization involved a spin-up process using 2001 meteorological forcing data until a quasiequilibrium state was reached, defined as inter-annual variations of less than 0.05 gC m<sup>-2</sup> yr<sup>-1</sup> in the slowest pools. Following initialization, we conducted transient simulations from 2002 to 2021 using the tuned parameter set. To evaluate model performance, we compared pool sizes from different nutrient coupling configurations (C-only, CN, and CNP) in 2021 with observed data (Tables S1-S3), assuming that our mature forest study site was at a quasi-steady state, where interannual changes in major pool sizes were negligible. The configuration that produced pool sizes closest to observations was selected to determine the initial state for subsequent simulations. Model performance was further evaluated by comparing simulated carbon fluxes in 2021 against observational data using both manually tuned and optimized parameters. The model evaluation metrics for carbon fluxes included the Root Mean Square Error (RMSE) and concordance correlation coefficient (CC), which quantify the absolute errors and the agreement between simulated and observed values. All statistical analyses and data visualizations were implemented in R (version 4.3.1).

#### 3 Results & Discussion

# 3.1 Evaluate the carbon-nutrient configurations

# 3.1.1 Carbon cycle

The CNP configuration accurately reproduced carbon pool sizes across ecosystem components. In contrast, the C-only and CN configurations tended to overestimate these pools (Fig. 4, Fig. 5a). In this P-limited site, the introduction of phosphorus limitations in CNP configurations progressively reduced carbon pool sizes compared to the C-only and CN configurations (Fig. 4a). This reduction reflects a fundamental assumption in carbon-nutrient coupled models that nutrient availability constrains carbon sequestration (Wieder et al., 2015; Sun et al., 2017) through various physiological processes (Jiang et al., 2019). At the ecosystem level (Fig. 4b), the C-only and CN configurations substantially overestimated total carbon stocks by 73.7% and 57.5%, respectively. In contrast, the CNP configuration produced estimates that were much closer to the observed values, with only a slight overestimate of 1.9%. The partitioning between plant and soil pools (Fig. 4b) showed that this overestimation occurred in both compartments, with the CNP configuration providing the closest match to observations.

Figure 4. Comparison of carbon pools among different nutrient coupling configurations. (a) Trajectories of ecosystem carbon pools during model spin-up for carbon-only (C-only), coupled carbon-nitrogen (CN), and coupled carbon-nitrogen-

phosphorus (CNP) simulations. The ecosystem carbon pool comprises nine pools within the components of plant, litter, and soil organic matter. (b) Comparison of simulated and observed (OBS) carbon pools in plant biomass and soil organic matter. Plant carbon pools comprise leaf, wood, and root carbon (excluding reproductive organs due to data unavailability), and soil carbon pools include fast, slow, and passive soil organic carbon components. The error bar for observation represents the standard deviation of the sum of plant and soil pools.

A more detailed examination of individual carbon pools (Fig. 5a) revealed that the overestimation was mainly contributed by wood and soil pools for C-only and CN configurations, which represent the major carbon stocks in the ecosystem. For plant components, wood carbon stocks were substantially overestimated by approximately 122.2% and 89.6% in the C-only and CN configurations, respectively. In contrast, the CNP configuration showed remarkable agreement with observations, with only a 5% deviation. Leaf carbon pools showed similar patterns of overestimation (C-only: 82.7%, CN: 59.1%, CNP: 3.6%). This improvement in leaf carbon estimation by CNP was further confirmed by better LAI prediction: the CNP configuration (3.94 m $^2$  m $^{-2}$ ) showed only 5% deviation from observations (3.75 ± 0.15 m $^2$  m $^{-2}$ ), while C-only and CN configurations overestimated by 85% and 61%, respectively.

The observed reduction in LAI represents a decrease in photosynthetic capacity achieved through nutrient limitation of plant growth, which reduces the photosynthetic leaf area rather than directly affecting leaf-level photosynthetic physiological parameters. The relationships between leaf nitrogen and phosphorus concentrations and photosynthetic traits (e.g.,  $V_{cmax}$ ,  $J_{max}$ ) are well established (Walker et al., 2014; Ellsworth et al., 2022) and have been incorporated into some land surface models (e.g., JULES-CNP). However, these large-scale emergent relationships significantly overestimated photosynthetic parameters at our study site (Table S5). At the same time, our site-specific dataset was insufficient to derive robust empirical relationships between nutrient concentrations and photosynthetic capacity. Future studies with more comprehensive site-level measurements could enhance this aspect of the model to represent nutrient-carbon interactions better.

Additionally, the CNP configuration better captured observed carbon fluxes compared to the C-only and CN configurations (Table 5). Although the total plant carbon litterfall rate was moderately overestimated by 22.7%, this still reflects improved simulation of aboveground carbon dynamics and could be further refined by incorporating reproductive pool measurements in future studies.

In contrast, root carbon pools showed an overestimation across all configurations, with CNP exhibiting the lowest bias (34.2%) and falling within one standard deviation of the observed values (Table S1), while the C-only and CN configurations showed larger deviations (68.8% and 65.1%, respectively). The relatively higher root carbon estimation in CNP may be attributed to its dynamic allocation strategy, which preferentially allocates carbon to roots under nutrient-limited conditions. While our model successfully reproduced the enhanced belowground carbon allocation under nutrient limitation, consistent with experimental evidence (Wu et al., 2025; Gill et al., 2016), the overestimated root carbon suggests additional constraints are needed. Indeed, the nutrient-dependent allocation scheme remains a significant source of uncertainty in terrestrial biosphere models (Zaehle et al., 2014; Jiang et al., 2024a). Although dynamic allocation schemes have been demonstrated to be significantly influenced by nutrient availability (Xia et al., 2023), explicit nutrient controls on allocation remain underrepresented in many ecosystem models (De Kauwe et al., 2014; but see Knox et al., 2024). Our model presents a practical approach for representing the nutrient regulation of carbon allocation processes. These results highlight the necessity of improved observational constraints on root turnover and carbon allocation patterns for more accurate process-based simulations.

Figure 5. Comparison of simulated and observed ecosystem pools across different nutrient coupling configurations (C-only, CN, and CNP). (a) Carbon pools in vegetation components (leaf, wood, root) and soil, with values for leaf and root scaled by  $10^1$ . (b) Nitrogen pools in vegetation components, soil (scaled by  $10^{-2}$ ), and mineral nitrogen (N<sub>min</sub>). (c) Phosphorus pools in vegetation components, soil organic P (scaled by  $10^{-2}$ ), labile P (P<sub>lab</sub>), and sorbed P (P<sub>s</sub>). Error bars on observed data (OBS) indicate standard deviations. Numbers in parentheses indicate scaling factors applied to improve visualization. For example, the soil P value marked with  $10^{-2}$  indicates that this value has been scaled down, and the actual value is  $1.58/10^{-2} = 158$  g P m<sup>-2</sup>. Shaded areas indicate inorganic nutrient pools.

For soil carbon pools, while C-only and CN configurations showed significant overestimations of 59.1% and 52.1%, respectively, the CNP configuration demonstrated the closest agreement with observations, with a slight overestimation of 1.06%. Despite the considerable observational uncertainty in soil carbon stocks (Table S1), the substantial overestimation by C-only and CN configurations was clearly beyond the reasonable range. This distinct improvement in soil carbon estimation by CNP configuration suggests that proper representation of nutrient limitations is crucial for realistic soil carbon predictions (Cui et al., 2024; Wei et al., 2022; Achat et al., 2016). In conclusion, the CNP model

consistently shows better alignment with observed carbon pools, particularly in reducing the systematic overestimation seen in the C-only and CN models.

**Table 5.** Observed and simulated carbon, nitrogen, and phosphorus fluxes with C-only, CN, and CNP configurations. The plant litterfall rate is the sum of the litterfall of leaves, woody, and reproductive parts.

| C, N and P fluxes               | C-only | CN    | CNP  | Observation         | Unit                                  |
|---------------------------------|--------|-------|------|---------------------|---------------------------------------|
| C transfer from leaf to litter  | 0.43   | 0.38  | 0.25 | $0.26\pm0.06$       | kg C m <sup>-2</sup> yr <sup>-1</sup> |
| C transfer from plant to litter | 0.98   | 0.86  | 0.54 | $0.44{\pm}0.04$     | kg C m <sup>-2</sup> yr <sup>-1</sup> |
| N transfer from plant to litter | -      | 11.36 | 7.44 | $6.74 \pm 0.68$     | g N m $^{-2}$ yr $^{-1}$              |
| P transfer from plant to litter | -      | -     | 0.24 | $0.79\pm0.24$       | g P $m^{-2}$ yr <sup>-1</sup>         |
| Soil respiration                | 1.72   | 1.59  | 1.13 | $0.99 \pm 0.07$     | kg C m <sup>-2</sup> yr <sup>-1</sup> |
| Net N mineralization            | -      | 18    | 12.3 | 13.14±0.73          | g N m <sup>-2</sup> yr <sup>-1</sup>  |
| Net P mineralization            | -      | -     | 0.54 | $0.67 \pm 0.14^{a}$ | g P m <sup>-2</sup> yr <sup>-1</sup>  |

<sup>a</sup> Jiang et al. (2024b).

# 3.1.2 N cycle

For nitrogen cycling properties, the CNP configuration exhibited superior performance in simulating nutrient pools compared to CN configurations (Fig. 5b). Regarding plant nitrogen pools, the CN configuration demonstrated substantial overestimations for leaf (59.2%), woody tissue (89.9%), and root N (55.9%). In contrast, the CNP configuration showed markedly improved accuracy, with only slight overestimations of 3.3%, 5.0% for leaf and wood N, and 28.8% for root N. The patterns of plant organic N across model configuration simulations were consistent with the carbon simulation results in both CN and CNP configurations, reflecting the constraints of plant tissue stoichiometry on coupled C-nutrient dynamics (Knox et al., 2024; Wang et al., 2010). For soil N pools, the CNP simulation (16.74 g N m<sup>-2</sup>) fell within the range of observed values (18.6  $\pm$  5.5 g N m<sup>-2</sup>), whereas the CN configuration substantially overestimated soil N (28.75 g N m<sup>-2</sup>). The slight underestimation of soil N in CNP relative to observations may be attributed to the flexible soil C:N ratios, as these ratios can vary within specific ranges due to complex microbial processes and dynamics of organic matter decomposition (Tian et al., 2010, 2021). The introduction of P cycling into the model resulted in reduced carbon allocation

to both plant and soil pools, which consequently led to proportional reductions in organic N pools compared to the CN configuration, ultimately capturing the observed N pools more accurately.

For soil mineral N content, the CN configuration underestimated soil mineral N content by 33.3% despite simulating higher net N mineralization rates (Table 5). This discrepancy likely reflects the absence of phosphorus constraints in the CN model. While the CN model simulated higher net N mineralization than the CNP model (Table 5), this enhanced nitrogen input was offset by excessive plant N uptake. This is consistent with the substantial overestimation of plant carbon pools in the CN configuration (Fig. 5a) and the correspondingly lower soil mineral N reserves (Fig. 5b). In contrast, the CNP configuration showed a moderate overestimation (15.9%) of soil mineral N content, demonstrating better agreement with observations compared to CN. The elevated soil mineral N levels in CNP could be attributed to the higher plant N litterfall rates (10.4% above observed rates, Table 5), which compensated for the underestimated net N mineralization rates.

The incorporation of P cycling constraints in the CNP configuration substantially improved the simulation of N pools and fluxes compared to the CN configuration, demonstrating the importance of considering N-P interactions in ecosystem modeling. This improvement reflects the fundamental interconnectedness of nitrogen and phosphorus cycles, where phosphorus availability directly regulates plant nitrogen demand and uptake efficiency, while nitrogen status influences phosphorus acquisition strategies (Elser et al., 2007; Peñuelas et al., 2013). In our model, these interactions are primarily captured through the tight coupling between soil nutrient availability, plant stoichiometry, and plant growth processes, which prevents unrealistic carbon and nitrogen accumulation when phosphorus becomes limiting. Notably, our model has limitations in capturing the full complexity of N-P interactions, reflecting broader challenges in coupled CNP modeling (Achat et al., 2016). For example, the absence of linkages between nitrogen fixation processes and phosphatase enzyme activity (Batterman et al., 2018), as well as the simplified representation of plant-microbe competition for nutrients and the lack of explicit mycorrhizal associations, suggest areas for future model refinement (Wu et al., 2023; Braghiere et al., 2021; Zhu et al., 2019).

## **3.1.3 P** cycle

varying environmental conditions.

The CNP model showed good overall performance in simulating phosphorus pools across ecosystem compartments (Fig. 5c). For plant components, the model accurately reproduced organic P pools, with slight overestimations of 5.0%, 2.8%, and 10.0% for leaf, wood, and root compartments, respectively. For the soil P, the CNP simulated a lower value (1.58 g P m<sup>-2</sup>) than observed, but within its range (1.8  $\pm$  0.6). Those organic P pools have the same pattern as organic N pools for CNP simulations, as C-N-P is coupled through stoichiometry. The simulated inorganic P content (0.8 g P m<sup>-2</sup>) fell within the observed range (0.48-1.6 g P m<sup>-2</sup>). Additionally, the simulated net P mineralization rate (0.54 g P m<sup>-2</sup> yr<sup>-1</sup>) was comparable to observations from tropical forests (0.67  $\pm$  0.14 g P m<sup>-2</sup> yr<sup>-1</sup>; Jiang et al., 2024b). The model successfully reproduced the observed levels of various P pools overall; however, it significantly underestimated plant P litterfall rates by 69% (Table 5). This discrepancy suggests potential limitations in the model's representation of nutrient-related processes, such as plant nutrient resorption mechanisms. Nutrient resorption is a crucial physiological process through which plants adapt to varying N and P availability in ecosystems. In our model, we implemented a fixed resorption coefficient (Table 4), which may oversimplify the dynamic nature of nutrient resorption. Additionally, our model does not account for the reciprocal effects of nitrogen and phosphorus availability on nutrient resorption dynamics, where N availability influences P resorption efficiency and vice versa (See et al., 2015; Li et al., 2019). This simplified representation likely contributes to the contrasting patterns observed in plant nutrient litterfall rates, which overestimate N litterfall while underestimating P litterfall. Plants typically adjust their nutrient resorption efficiency in response to both internal nutrient status and external resource availability (Mao et al., 2015; Sasha et al., 2012; Aerts and Chapin, 2000; Aerts, 1996). The fixed resorption coefficients in the current model structure may not capture these adaptive responses, potentially leading to unrealistic nutrient cycling patterns, especially under

The CNP configuration successfully captured the steady-state P distributions across ecosystem pools despite some discrepancies in P cycling processes. Further refinements in P cycling processes, particularly in plant-soil P transfer mechanisms and plant internal P

recycling, would be valuable for improving model performance (Jiang et al., 2019; 2024a). However, these improvements are currently constrained by limited observational data, as data scarcity remains a significant challenge for C-nutrient coupled modeling (Achat et al., 2016; Reed et al., 2015). Future research should prioritize comprehensive field measurements of P cycling processes, including plant P resorption efficiency, soil P transformation rates, and plant-soil P transfer dynamics. Such empirical data would not only help validate and improve model performance but also enhance our understanding of terrestrial P cycling and its interactions with C and N cycles in terrestrial ecosystems.

### 3.2 Evaluate the model-data fusion module

To evaluate the efficiency of the integrated data assimilation module, we compared the carbon fluxes from CNP simulations with default and optimized parameters (Figs. 6 and 7). The optimization showed varied improvements across different carbon flux components. For gross primary productivity (GPP), both default and optimized simulations captured the seasonal patterns well, with only a minor improvement in RMSE from 10.94 to 10.69 and a slightly increased correlation coefficient from 0.53 to 0.57 after optimization (Fig. 6a, e).

The photosynthetic capacity per unit area and photosynthetic surface area, indicated by  $V_{cmax}$  and SLA respectively, are key determinants of GPP. Both  $V_{cmax}$  and SLA were adjusted within their reference ranges during data assimilation (Fig. 8). Although these parameters showed compensatory effects in their adjustments, their combined effect still demonstrated a tendency to enhance GPP (Fig. 6a, e). Notably, the systematic underestimation of GPP, particularly during the growing season, suggests the need for improving current carbon cycle process representations. These improvements should include (1) the soil moisture control on stomatal conductance specific to evergreen broadleaf forests (Weng & Luo, 2008) and (2) the calculation of sunlit and shaded leaf proportions through more accurate clumping index parameterization in the two-leaf model (Wang et al., 2024; Bi et al., 2022; Yan et al., 2017).

Figure 6. Comparison of weekly observed and simulated carbon fluxes using default parameters and optimized parameters for the Tiantong site in 2021. (a-c) Time series of observed (black dots) and simulated values with default parameters (blue line) and optimized parameters (red line), where the optimized results are derived from 1000 parameter sets randomly selected from 10,000 accepted parameter sets during the data assimilation process (shaded areas represent standard deviation). (d-f) Scatter plots of simulated versus observed values corresponding to the time series above, where the dashed line represents the 1:1 line. CC, correlation coefficient; RMSE, root mean square error.

Figure 7. Diurnal patterns of hourly net ecosystem exchange (NEE) across different months simulated by the CNP model configuration before (default) and after data assimilation (MCMC) compared with observations. Black lines represent observational data with shaded areas indicating  $\pm$  1 standard deviation (SD). Colored lines indicate model simulations with shaded areas showing their respective  $\pm$  1 SD. Root mean square errors (RMSE) between model outputs and observations are colored in blue for simulations with default parameters and in red for simulations with accepted parameters.

Ecosystem respiration (ER) showed more substantial improvement with data assimilation, with RMSE decreasing from 11.03 to 6.72 g C per m² per week, particularly in reducing the high-frequency fluctuations present in the default simulation (Fig. 6b). This improvement in ER led to a notable improvement in NEE, where the RMSE decreased from 14.21 to 8.83 g C per m² per week, and the correlation coefficient improved dramatically from -0.03 to 0.51. The significantly improved representation of carbon exchange dynamics with parameter optimization is further confirmed by the diurnal patterns across months (Fig. 7), with reduced RMSE in most months (7 out of 12). However, certain limitations persist, notably the underestimated NEE during midday hours in the growing season, primarily attributed to underestimated GPP, which requires further investigation.

The enhancement in ER and NEE primarily resulted from the efficiently constrained key parameters (Table 6, Fig. 8) based on the validated state variables (Fig. 5). While the default parameters achieved reasonable state variables, the response of state variables to new meteorological forcing conditions required adjustment (Ma et al., 2021). For instance, the Q<sub>10</sub> and soil carbon residence time (T<sub>6</sub>-T<sub>8</sub>) are well-constrained in our case. The temperature sensitivity parameter represents microbial responses to soil temperature, and carbon residence times serve as a proxy for microbial accessibility to carbon substrates, rather than just soil carbon properties, both of which are related to heterotrophic respiration. Through the optimization of these parameters, the CNP model effectively reduced the high-frequency fluctuations present in the default simulation and better captured the observed temporal dynamics.

Data assimilation substantially improved CNP model performance in carbon flux simulation, highlighting the potential for applying our developed model to other flux sites without tedious manual calibration procedures. Given that parameter optimization can potentially compensate for structural deficiencies in models (e.g., the equifinality issue; Luo et al., 2016, 2009; Sierra et al., 2015), it's understandable that models with different nutrient coupling schemes can generate similar performance with optimized parameters (Fig. S1, Text S1). However, while parameter optimization can help the C-only model fit historical data, it may result in unrealistic parameter values (Fig. S2) and essentially "bakes

in" current nutrient conditions without representing the underlying processes, thereby compromising its predictive capacity for future scenarios.

Figure 8. Posterior distributions of model parameters derived from Bayesian calibration. Grey shaded areas represent parameter posterior distributions, with red and blue vertical lines indicating posterior means and default values, respectively. The parameters (listed in Table 6) include  $Q_{10}$ , SLA,  $V_{cmax}$ , and carbon residence time parameters ( $T_1$ - $T_9$ ). The corresponding numerical values are shown in matching colors.

**Table 6.** Target parameters, their ranges, mean values and maximum likelihood estimation (MLE) of the posterior distribution.  $Q_{10}$  represents temperature sensitivity; SLA, specific leaf area; and  $V_{cmax}$ , maximum carboxylation rate.  $T_1$ – $T_9$  indicate turnover times for individual pools.

| Parameters           | Lower  | Upper   | Mean    | MLE     |  |  |  |  |
|----------------------|--------|---------|---------|---------|--|--|--|--|
| Q <sub>10</sub>      | 1.00   | 3.00    | 1.29    | 1.26    |  |  |  |  |
| SLA                  | 89.04  | 184.26  | 147.23  | 166.68  |  |  |  |  |
| Vcmax                | 23.29  | 29.11   | 24.52   | 24.42   |  |  |  |  |
| Carbon turnover rate |        |         |         |         |  |  |  |  |
| $T_1$                | 0.25   | 8.76    | 5.19    | 6.11    |  |  |  |  |
| $T_2$                | 25.00  | 750.00  | 373.13  | 260.58  |  |  |  |  |
| T <sub>3</sub>       | 0.24   | 1.80    | 1.03    | 0.79    |  |  |  |  |
| $T_4$                | 0.10   | 5.00    | 2.19    | 0.76    |  |  |  |  |
| T <sub>5</sub>       | 0.10   | 0.50    | 0.27    | 0.21    |  |  |  |  |
| $T_6$                | 0.50   | 20.00   | 7.75    | 1.69    |  |  |  |  |
| $T_7$                | 0.05   | 1.00    | 0.53    | 0.43    |  |  |  |  |
| T <sub>8</sub>       | 2.00   | 200.00  | 26.41   | 9.75    |  |  |  |  |
| Т9                   | 400.00 | 2000.00 | 1197.29 | 1090.48 |  |  |  |  |

### 4 Conclusions

In this study, we developed and evaluated a process-based CNP-coupled model for subtropical evergreen broadleaf forest. The CNP configuration demonstrated superior performance compared to C-only and CN models across most biogeochemical pools and fluxes, effectively addressing the overestimation issues prevalent in models with simplified biogeochemical processes. The incorporation of phosphorus cycling mechanisms proved crucial for capturing ecosystem dynamics in these phosphorus-limited systems, providing an essential foundation for predicting subtropical evergreen broadleaf forest responses to climate change. Beyond mechanistic improvements, site-scale models like TECO-CNP can fully leverage rich, localized datasets, including forest inventory records, experimental manipulations, and eddy covariance measurements, to constrain model parameters and processes. This integration is crucial because unobserved or weakly observed processes cannot be reliably constrained through data assimilation alone (Luo et al., 2011). TECO-

CNP is designed to facilitate the fusion of such multi-process information, thereby enabling more mechanistic and robust representations of ecosystem C-N-P dynamics. Furthermore, we implemented and evaluated a model-data fusion framework using the MCMC algorithm, which significantly improved the simulation of carbon fluxes. The optimization of key parameters, including those that control photosynthetic capacity, temperature sensitivity, and carbon turnover rate, effectively reduced simulation uncertainties and enhanced model performance. The success of the data assimilation approach not only demonstrates its effectiveness in current model optimization but also provides a promising path for future model improvement and applications across diverse ecosystems. More importantly, integrating data assimilation frameworks with site-level biogeochemical models facilitates a synergistic loop between experimental findings and model development, enhancing our understanding of the nutrient cycle processes and our ability to make reliable predictions. This integrated approach provides a robust framework for improving ecosystem models and advancing our understanding of nutrient cycling in response to environmental changes.

1009

# Code availability

- The model code is available at https://doi.org/10.5281/zenodo.15032706, (Wan, 2025a).
- Data availability
- The model outputs related to the results in this paper are provided in a Zenodo repository
- (https://doi.org/10.5281/zenodo.15033861, Wan, 2025b). The visualization scripts and
- associated data for generating all figures are provided in a separate Zenodo repository
- (https://doi.org/10.5281/zenodo.15032690, Wan, 2025c).

### 1007 Supplement link

The supplementary material is published with this article.

### **Author contribution**

- J.X. and F.W. conceived and designed the study. F.W. developed the model, implemented
- the code, and performed the analysis. C.B. provided technical support. K.H., C.B., and F.W.
- prepared the forcing data and eddy flux data. Y.L. and E.W. provided valuable suggestions

for manuscript content and improvement. F.W. wrote the first draft of the manuscript, and all authors reviewed and approved the final version.

Competing interests

The authors declare that they have no conflict of interest.

# 1018 Acknowledgments

This work is financially supported by the National Natural Science Foundation of China (31722009), the National Key R&D Program of China (2022YFF0802104), and the Shanghai Pilot Program for Basic Research (TQ20220102). We thank Zemei Zheng, Erqian Cui, and Yang Qiao for maintaining the field measurements and providing the observational data. We also thank Cuihai You for their assistance in eddy covariance flux data collection and processing.

- Achat, D. L., Augusto, L., Gallet-Budynek, A., and Loustau, D.: Future challenges in coupled C–N–P cycle models for terrestrial ecosystems under global change: a review, Biogeochemistry, 131, 173-202, <a href="https://doi.org/10.1007/s10533-016-0274-9">https://doi.org/10.1007/s10533-016-0274-9</a>, 2016.
- Aerts R., Nutrient resorption from senescing leaves of perennials: are there general patterns?

  Journal of Ecology 84, 597–608, <a href="https://doi.org/10.2307/2261481">https://doi.org/10.2307/2261481</a>, 1996.
- Arora, V. K., and Boer, G. J.: A parameterization of leaf phenology for the terrestrial ecosystem component of climate models, Global Change Biol., 11, 39-59, https://doi.org/10.1111/j.1365-2486.2004.00890.x, 2005.
- Ball, J. T., Woodrow, I. E., and Berry, J. A.: A model predicting stomatal conductance and its contribution to the control of photosynthesis under different environmental conditions, in: Progress in Photosynthesis Research, Vol. IV, edited by: Biggens, J., Martinus Nijhoff Publishers, Dordrecht, Netherlands, 221-224, <a href="https://doi.org/10.1007/978-94-017-0519-6">https://doi.org/10.1007/978-94-017-0519-6</a> 48, 1987.
- Barraclough, P. B., and Tinker, P. B.: The determination of ionic diffusion coefficients in field soils. I. Diffusion coefficients in sieved soils in relation to water content and bulk density, J. Soil Sci., 32, 225-236, <a href="https://doi.org/10.1111/j.1365-2389.1981.tb01702.x">https://doi.org/10.1111/j.1365-2389.1981.tb01702.x</a>, 1981.
- Barrow, N.: The description of sorption curves, Eur. J. Soil Sci., 59, 900-910, 1045 <a href="https://doi.org/10.1111/j.1365-2389.2008.01041.x">https://doi.org/10.1111/j.1365-2389.2008.01041.x</a>, 2008.
- Batterman, S. A., Hall, J. S., Turner, B. L., Hedin, L. O., LaHaela Walter, J. K., Sheldon, P., and Van Breugel, M.: Phosphatase activity and nitrogen fixation reflect species differences, not nutrient trading or nutrient balance, across tropical rainforest trees, Ecol. Lett., 21, 1486-1495, https://doi.org/10.1111/ele.13129, 2018.
- Ben Keane, J., Hartley, I. P., Taylor, C. R., Leake, J. R., Hoosbeek, M. R., Miglietta, F., and Phoenix, G. K.: Grassland responses to elevated CO<sub>2</sub> determined by plant-microbe competition for phosphorus, Nat. Geosci., 16, 704-709, <a href="https://doi.org/10.1038/s41561-023-01225-z">https://doi.org/10.1038/s41561-023-01225-z</a>, 2023.
- Bi, W., He, W., Zhou, Y., and Yang, F.: A global 0.05° dataset for gross primary production of sunlit and shaded vegetation canopies from 1992 to 2020, Scientific Data, 9, 213, <a href="https://doi.org/10.1038/s41597-022-01309-2">https://doi.org/10.1038/s41597-022-01309-2</a>, 2022.
- Bieleski, R. L.: Phosphate Pools, Phosphate Transport, and Phosphate Availability, Annu.

  Rev. Plant Physio., 24, 225-252,

  https://doi.org/10.1146/annurev.pp.24.060173.001301, 1973.
- Binkley, D., Smith, F. W., and Son, Y.: Nutrient supply and declines in leaf area and production in lodgepole pine, Can. J. Forest Res., 25, 621-628, <a href="https://doi.org/10.1139/x95-069">https://doi.org/10.1139/x95-069</a>, 1995.
- Chapin, F. S. III: The mineral nutrition of wild plants, Annu. Rev. Ecol. Syst., 11, 233-260, 1064 <a href="https://doi.org/10.1146/annurev.es.11.110180.001313">https://doi.org/10.1146/annurev.es.11.110180.001313</a>, 1980.
- Chou, C. B., Hedin, L. O., and Pacala, S. W.: Functional groups, species and light interact with nutrient limitation during tropical rainforest sapling bottleneck, J. Ecol., 106, 157-167, <a href="https://doi.org/10.1111/1365-2745.12823">https://doi.org/10.1111/1365-2745.12823</a>, 2018.
- Cui, E., Lu, R., Xu, X., Sun, H., Qiao, Y., Ping, J., Qiu, S., Lin, Y., Bao, J., Yong, Y., Zheng, Z., Yan, E., and Xia, J.: Soil phosphorus drives plant trait variations in a

- mature subtropical forest, Glob. Change Biol., 28, 3310-3320, 1071 <a href="https://doi.org/10.1111/gcb.16148">https://doi.org/10.1111/gcb.16148</a>, 2022.
- Cui, Y., Hu, J., Peng, S., Delgado-Baquerizo, M., Moorhead, D. L., Sinsabaugh, R. L., Xu, X., Geyer, K. M., Fang, L., Smith, P., Peñuelas, J., Kuzyakov, Y., Chen, J., Limiting resources define the global pattern of soil microbial carbon use efficiency. Adv. Sci., 11, 2308176, https://doi.org/10.1002/advs.202308176, 2024.
- Cunha, H. F. V., Andersen, K. M., Lugli, L. F., Santana, F. D., Aleixo, I. F., Moraes, A.
  M., Garcia, S., Di Ponzio, R., Mendoza, E. O., Brum, B., Rosa, J. S., Cordeiro, A. L.,
  Portela, B. T. T., Ribeiro, G., Coelho, S. D., De Souza, S. T., Silva, L. S., Antonieto,
  F., Pires, M., and Quesada, C. A.: Direct evidence for phosphorus limitation on
  Amazon forest productivity, Nature, 608, 558-562, <a href="https://doi.org/10.1038/s41586-022-05085-2">https://doi.org/10.1038/s41586-022-05085-2</a>, 2022.
- De Kauwe, M. G., Medlyn, B. E., Zaehle, S., Walker, A. P., Dietze, M. C., Wang, Y., Luo, Y., Jain, A. K., El-Masri, B., Hickler, T., Wårlind, D., Weng, E., Parton, W. J., Thornton, P. E., Wang, S., Prentice, I. C., Asao, S., Smith, B., McCarthy, H. R., and Norby, R. J.: Where does the carbon go? A model-data intercomparison of vegetation carbon allocation and turnover processes at two temperate forest free-air CO<sub>2</sub> enrichment sites, New Phytol., 203, 883-899, <a href="https://doi.org/10.1111/nph.12847">https://doi.org/10.1111/nph.12847</a>, 2014.
- Denison, F., and Loomis, B.: An integrative physiological model of Alfalfa growth and development, UC ANR Publication 1926, Univ. California, Davis, 1989.
- Du, E., de Vries, W., Han, W., Liu, X., Yan, Z., and Jiang, Y.: Imbalanced phosphorus and nitrogen deposition in China's forests, Atmos. Chem. Phys., 16, 8571-8579, <a href="https://doi.org/10.5194/acp-16-8571-2016">https://doi.org/10.5194/acp-16-8571-2016</a>, 2016.
- Du, E., Terrer, C., Pellegrini, A. F. A., Ahlström, A., van Lissa, C. J., Zhao, X., Xia, N., Wu, X., and Jackson, R. B.: Global patterns of terrestrial nitrogen and phosphorus limitation, Nat. Geosci., 13, 221-226, <a href="https://doi.org/10.1038/s41561-019-0530-4">https://doi.org/10.1038/s41561-019-0530-4</a>, 2020.
- Du, Z., Weng, E., Jiang, L., Luo, Y., Xia, J., and Zhou, X.: Carbon-nitrogen coupling under three schemes of model representation: A traceability analysis, Geosci. Model Dev., 11, 4399-4416, <a href="https://doi.org/10.5194/gmd-11-4399-2018">https://doi.org/10.5194/gmd-11-4399-2018</a>, 2018.
- Du, Z., Wang, J., Zhou, G., Bai, S. H., Zhou, L., Fu, Y., Wang, C., Wang, H., Yu, G., & Zhou, X.: Differential effects of nitrogen vs. Phosphorus limitation on terrestrial carbon storage in two subtropical forests: A Bayesian approach. Science of The Total Environment, 795, 148485, <a href="https://doi.org/10.1016/j.scitotenv.2021.148485">https://doi.org/10.1016/j.scitotenv.2021.148485</a>, 2021.
- Ellsworth, D. S., Crous, K. Y., De Kauwe, M. G., Verryckt, L. T., Goll, D., Zaehle, S., Bloomfield, K. J., Ciais, P., Cernusak, L. A., Domingues, T. F., Dusenge, M. E., Garcia, S., Guerrieri, R., Ishida, F. Y., Janssens, I. A., Kenzo, T., Ichie, T., Medlyn, B. E., Meir, P., and Wright, I. J.: Convergence in phosphorus constraints to photosynthesis in forests around the world, Nat. Commun., 13, 5005, https://doi.org/10.1038/s41467-022-32545-0, 2022.
- Elser, J. J., Bracken, M. E. S., Cleland, E. E., Gruner, D. S., Harpole, W. S., Hillebrand, H., Ngai, J. T., Seabloom, E. W., Shurin, J. B., and Smith, J. E.: Global analysis of nitrogen and phosphorus limitation of primary producers in freshwater, marine and

- terrestrial ecosystems, Ecol. Lett., 10, 1135-1142, <a href="https://doi.org/10.1111/j.1461-0248.2007.01113.x">https://doi.org/10.1111/j.1461-0248.2007.01113.x</a>, 2007.
- Farquhar, G. D., von Caemmerer, S., and Berry, J. A.: A biochemical model of photosynthetic CO<sub>2</sub> assimilation in leaves of C3 species, Planta, 149, 78-90, <a href="https://doi.org/10.1007/BF00386231">https://doi.org/10.1007/BF00386231</a>, 1980.
- Fisher, J. B., Badgley, G., and Blyth, E.: Global nutrient limitation in terrestrial vegetation, Global Biogeochem. Cy., 26, GB3007, <a href="https://doi.org/10.1029/2011GB004252">https://doi.org/10.1029/2011GB004252</a>, 2012.
- Fisher, R. A., and Koven, C. D.: Perspectives on the future of land surface models and the challenges of representing complex terrestrial systems, J. Adv. Model. Earth Syst., 12, e2018MS001453, https://doi.org/10.1029/2018MS001453, 2020.
- Fleischer, K., Rammig, A., De Kauwe, M. G., Walker, A. P., Domingues, T. F.,
  Fuchslueger, L., Garcia, S., Goll, D. S., Grandis, A., Jiang, M., Haverd, V., Hofhansl,
  F., Holm, J. A., Kruijt, B., Leung, F., Medlyn, B. E., Mercado, L. M., Norby, R. J.,
  Pak, B., von Randow, C., Quesada, C. A., Schaap, K. J., Valverde-Barrantes, O. J.,
  Wang, Y.-P., Yang, X., Zaehle, S., Zhu, Q., and Lapola, D. M.: Amazon forest
  response to CO2 fertilization dependent on plant phosphorus acquisition, Nat. Geosci.,
  12, 736-741, https://doi.org/10.1038/s41561-019-0404-9, 2019.
- Friedlingstein, P., O'Sullivan, M., Jones, M., Andrew, R. M., Bakker, D. C. E., Hauck, J., Landschützer, P., Le Quéré, C., Luijkx, I., Peters, G. P., Peters, W., Pongratz, J., Schwingshackl, C., Sitch, S., Canadell, J. G., Ciais, P., Jackson, R. B., Alin, S., Anthoni, P., and Zheng, B.: Global Carbon Budget 2023, Earth Syst. Sci. Data, 15, 5301-5369, https://doi.org/10.5194/essd-15-5301-2023, 2023.
- Gill, A.L. and Finzi, A.C., Belowground carbon flux links biogeochemical cycles and resource-use efficiency at the global scale. Ecol Lett, 19, 1419-1428. https://doi.org/10.1111/ele.12690, 2016.

1141

- Goll, D. S., Brovkin, V., Parida, B. R., Reick, C. H., Kattge, J., Reich, P. B., van Bodegom, P. M., and Niinemets, Ü.: Nutrient limitation reduces land carbon uptake in simulations with a model of combined carbon, nitrogen and phosphorus cycling, Biogeosciences, 9, 3547-3569, <a href="https://doi.org/10.5194/bg-9-3547-2012">https://doi.org/10.5194/bg-9-3547-2012</a>, 2012.
- Goll, D. S., Vuichard, N., Maignan, F., Jornet-Puig, A., Sardans, J., Violette, A., Peng, S., Sun, Y., Kvakic, M., Guimberteau, M., Guenet, B., Zaehle, S., Penuelas, J., Janssens, I., and Ciais, P.: A representation of the phosphorus cycle for ORCHIDEE (revision 4520), Geosci. Model Dev., 10, 3745-3770, <a href="https://doi.org/10.5194/gmd-10-3745-2017">https://doi.org/10.5194/gmd-10-3745-2017</a>, 2017.
- Grant, R. F., Black, T. A., den Hartog, G., Berry, J. A., Neumann, H. H., Blanken, P. D., Yang, P. C., Russell, C., and Nalder, I. A.: Diurnal and annual exchanges of mass and energy between an aspen-hazelnut forest and the atmosphere: Testing the mathematical model Ecosys with data from the BOREAS experiment, J. Geophys. Res.-Atmos., 104, 27699-27717, https://doi.org/10.1029/1998JD200117, 1999.
- Grassi, G., House, J., Dentener, F., Federici, S., den Elzen, M., and Penman, J.: The key role of forests in meeting climate targets requires science for credible mitigation, Nat. Clim. Change, 7, 220-226, <a href="https://doi.org/10.1038/nclimate3227">https://doi.org/10.1038/nclimate3227</a>, 2017.
- Gutschick, V. P.: Evolved Strategies in Nitrogen Acquisition by Plants, Am. Nat., 118, 607-637, https://doi.org/10.1086/283858, 1981.

- Haverd, V., Smith, B., Nieradzik, L., Briggs, P. R., Woodgate, W., Trudinger, C. M., Canadell, J. G., and Cuntz, M.: A new version of the CABLE land surface model (Subversion revision r4601) incorporating land use and land cover change, woody vegetation demography, and a novel optimisation-based approach to plant coordination of photosynthesis, Geosci. Model Dev., 11, 2995-3026, https://doi.org/10.5194/gmd-11-2995-2018, 2018.
- Hou, E., Lu, X., Jiang, L., Wen, D., and Luo, Y.: Quantifying Soil Phosphorus Dynamics:

  A Data Assimilation Approach, J. Geophys. Res.-Biogeo., 124, 2159-2173, https://doi.org/10.1029/2018JG004903, 2019.
- Hou, E., Luo, Y., Kuang, Y., Chen, C., Lu, X., Jiang, L., Luo, X., and Wen, D.: Global meta-analysis shows pervasive phosphorus limitation of aboveground plant production in natural terrestrial ecosystems, Nat. Commun., 11, 637, <a href="https://doi.org/10.1038/s41467-020-14492-w">https://doi.org/10.1038/s41467-020-14492-w</a>, 2020.
- Hou, E., Tan, X., Heenan, M., and Wen, D.: A global dataset of plant available and unavailable phosphorus in natural soils derived by Hedley method, Sci. Data, 5, 180166, https://doi.org/10.1038/sdata.2018.166, 2018.
- Hou, E., Wen, D., Jiang, L., Luo, X., Kuang, Y., Lu, X., Chen, C., Allen, K. T., He, X., Huang, X., and Luo, Y.: Latitudinal patterns of terrestrial phosphorus limitation over the globe, Ecol. Lett., 24, 1420-1431, <a href="https://doi.org/10.1111/ele.13761">https://doi.org/10.1111/ele.13761</a>, 2021.
- Houlton, B. Z., Wang, Y.-P., Vitousek, P. M., and Field, C. B.: A unifying framework for dinitrogen fixation in the terrestrial biosphere, Nature, 454, 327-330, https://doi.org/10.1038/nature07028, 2008.
- Huang W, Zhou G, Liu J, Duan H, Liu X, Fang X, Zhang D.: Shifts in soil phosphorus fractions under elevated CO<sub>2</sub> and N addition in model forest ecosystems in subtropical China. Plant Ecol., 215, 1373–1384, <a href="https://doi.org/10.1007/s11258-014-0394-z">https://doi.org/10.1007/s11258-014-0394-z</a>, 2014.
- Huang, Y., Jiang, J., Ma, S., Ricciuto, D., Hanson, P. J., and Luo, Y.: Soil thermal dynamics, snow cover, and frozen depth under five temperature treatments in an ombrotrophic bog: Constrained forecast with data assimilation, J. Geophys. Res.-Biogeo., 122, 2046-2063, https://doi.org/10.1002/2016JG003725, 2017.
- Jiang, L., Shi, Z., Xia, J., Liang, J., Lu, X., Wang, Y., and Luo, Y.: Transient Traceability
  Analysis of Land Carbon Storage Dynamics: Procedures and Its Application to Two
  Forest Ecosystems, J. Adv. Model. Earth Syst., 9, 2822-2835,

  https://doi.org/10.1002/2017MS001004, 2017.
- Jiang, M., Caldararu, S., Zaehle, S., Ellsworth, D. S., and Medlyn, B. E.: Towards a more physiological representation of vegetation phosphorus processes in land surface models, New Phytol., 222, 1223-1229, https://doi.org/10.1111/nph.15688, 2019.
- Jiang, M., Medlyn, B. E., Wårlind, D., Knauer, J., Fleischer, K., Goll, D. S., Olin, S., Yang,
  X., Yu, L., Zaehle, S., Zhang, H., Lv, H., Crous, K. Y., Carrillo, Y., Macdonald, C.,
  Anderson, I., Boer, M. M., Farrell, M., Gherlenda, A., and Smith, B.: Carbon-phosphorus cycle models overestimate CO2 enrichment response in a mature
  Eucalyptus forest, Sci. Adv., 10, eadl5822, <a href="https://doi.org/10.1126/sciadv.adl5822">https://doi.org/10.1126/sciadv.adl5822</a>,
  2024a.
- Jiang, M., Crous, K. Y., Carrillo, Y., Macdonald, C. A., Anderson, I. C., Boer, M. M.,
   Farrell, M., Gherlenda, A. N., Castañeda-Gómez, L., Hasegawa, S., Jarosch, K.,

- Milham, P. J., Ochoa-Hueso, R., Pathare, V., Pihlblad, J., Piñeiro, J., Powell, J. R., Power, S. A., Reich, P. B., and Ellsworth, D. S.: Microbial competition for phosphorus limits the CO2 response of a mature forest, Nature, 630, 660-665, https://doi.org/10.1038/s41586-024-07491-0, 2024b.
- Johnson, A. H., Frizano, J., Vann, D. R., and Johnson, R. A. H.: Biogeochemical implications of labile phosphorus in forest soils determined by the Hedley fractionation procedure, Oecologia, 135, 487-499, <a href="https://doi.org/10.1007/s00442-002-1164-5">https://doi.org/10.1007/s00442-002-1164-5</a>, 2003.
- Jonasson, S., Michelsen, A., Schmidt, I. K., and Nielsen, E. V.: Responses in microbes and plants to changed temperature, nutrient, and light regimes in the Arctic, Ecology, 80, 1828-1843, <a href="https://doi.org/10.1890/0012-9658(1999)080[1828:RIMAPT]2.0.CO;2">https://doi.org/10.1890/0012-9658(1999)080[1828:RIMAPT]2.0.CO;2</a>, 1999.
- Jones, S., Rowland, L., Cox, P., Hemming, D., Wiltshire, A., Williams, K., and Harper, A.
  B.: The impact of a simple representation of non-structural carbohydrates on the
  simulated response of tropical forests to drought, Biogeosciences, 17, 3589-3612,
  https://doi.org/10.5194/bg-17-3589-2020, 2020.
- Keenan, T. F., and Williams, C. A.: The Terrestrial Carbon Sink, Annu. Rev. Environ. 1220 Resour., 43, 219-243, https://doi.org/10.1146/annurev-environ-102017-030204, 2018.
- Keith, K., Raison, J. R., and Jacobsen, K. L.: Allocation of carbon in a mature eucalypt forest and some effects of soil phosphorus availability, Plant Soil, 196, 81-99, 1997.
- Knox, R. G., Koven, C. D., Riley, W. J., Walker, A. P., Wright, S. J., and Holm, J. A.:
  Nutrient dynamics in a coupled terrestrial biosphere and land model (ELM-FATES-CNP), J. Adv. Model. Earth Syst., 16, e2023MS003689, https://doi.org/10.1029/2023MS003689, 2024.
- Koerselman, W., and Mueleman, A. F. M.: The vegetation N:P ratio: a new tool to detect the nature of nutrient limitation, J. Appl. Ecol., 33, 1441-1450, https://doi.org/10.2307/2404783, 1996.
- Laliberté, E., Lambers, H., Burgess, T. I., and Wright, S. J.: Phosphorus limitation, soil-borne pathogens and the coexistence of plant species in hyperdiverse forests and shrublands, New Phytol., 206, 507-521, <a href="https://doi.org/10.1111/nph.13203">https://doi.org/10.1111/nph.13203</a>, 2015.
- Lambers, H., Shane, M. W., Cramer, M. D., Pearse, S. J., and Veneklaas, E. J.: Root structure and functioning for efficient acquisition of phosphorus: matching morphological and physiological traits, Ann. Bot.-London, 98, 693-713, <a href="https://doi.org/10.1111/nph.15833">https://doi.org/10.1111/nph.15833</a>, 2006.
- Leuning, R.: A critical appraisal of a combined stomatal-photosynthesis model for C3 plants, Plant Cell Environ., 18, 339-355, <a href="https://doi.org/10.1111/j.1365-3040.1995.tb00370.x">https://doi.org/10.1111/j.1365-3040.1995.tb00370.x</a>, 1995.
- Li, G.Y., Yang, X. D., Shi, Q. R., Ma, W. J., Wang, X. H., and Yan, E. R.: Effects of clear-felling on soil nutrient pools and nitrogen mineralization and nitrification in Tiantong, Zhejiang Province, Chinese Journal of Ecology, 33, 709-715, 10.13292/j.1000-4890.2014.0062, 2014.
- Li, L., Liu, B., Gao, X., Zhang, M., Liu, M., Zhang, B., Zhang, H., Zhao, Y., Xiao, C., and Zhou, J.: Nitrogen and phosphorus addition differentially affect plant ecological stoichiometry in desert grassland, Sci. Rep., 9, 18673, https://doi.org/10.1038/s41598-019-55275-8, 2019.

- Liang, G., Luo, Y., Zhou, Z., and Waring, B. G.: Nitrogen effects on plant productivity change at decadal time-scales, Global Ecol. Biogeogr., 30, 2488-2499, https://doi.org/10.1111/geb.13391, 2021.
- Linder, S., and Rook, D. A.: Effects of mineral nutrition on the carbon dioxide exchange of trees, in: Nutrition of forest trees in plantations, edited by: Bowen, G. D. and Nambiar, E. K. S., Academic Press, London, 211-236, 1984.
- Lu, C., and Tian, H.: Global nitrogen and phosphorus fertilizer use for agriculture production in the past half century: Shifted hot spots and nutrient imbalance, Earth Syst. Sci. Data, 9, 181-192, <a href="https://doi.org/10.5194/essd-9-181-2017">https://doi.org/10.5194/essd-9-181-2017</a>, 2017.
- Luo, M., Moorhead, D. L., Ochoa-Hueso, R., Mueller, C. W., Ying, S. C., and Chen, J.: Nitrogen loading enhances phosphorus limitation in terrestrial ecosystems with 1258 1259 implications for soil carbon cycling, Funct. Ecol., 36, 2845-2858, 1260 https://doi.org/10.1111/1365-2435.14178, 2022.
- Luo, Y., Ahlström, A., Allison, S. D., Batjes, N. H., Brovkin, V., Carvalhais, N., Chappell, A., Ciais, P., Davidson, E. A., Finzi, A., Georgiou, K., Guenet, B., Hararuk, O., 1262 1263 Harden, J. W., He, Y., Hopkins, F., Jiang, L., Koven, C., Jackson, R. B., Jones, C. D., 1264 Lara, M. J., Liang, J., McGuire, A. D., Parton, W., Peng, C., Randerson, J. T., Salazar, 1265 A., Sierra, C. A., Smith, M. J., Tian, H., Todd-Brown, K. E. O., Torn, M., van Groenigen, K. J., Wang, Y. P., West, T. O., Wei, Y., Wieder, W. R., Xia, J., Xu, X., 1266 1267 Xu, X., and Zhou, T.: Toward more realistic projections of soil carbon dynamics by 1268 Earth system models, Global Biogeochem. Cy., 30, 40-56, https://doi.org/10.1002/2015GB005239, 2016. 1269
- Luo, Y., and Reynolds, J. F.: Validity of extrapolating field CO2 experiments to predict carbon sequestration in natural ecosystems, Ecology, 80, 1568-1583, https://doi.org/10.1890/0012-9658(1999)080[1568:VOEFCE]2.0.CO;2, 1999.
- Luo, Y., Meyerhoff, P. A., and Loomis, R. S.: Seasonal patterns and vertical distributions of fine roots of alfalfa (*Medicago sativa L.*), Field Crop. Res., 40, 119-127, <a href="https://doi.org/10.1016/0378-4290(94)00090-Y">https://doi.org/10.1016/0378-4290(94)00090-Y</a>, 1995.
- Luo, Y., Ogle, K., Tucker, C., Fei, S., Gao, C., LaDeau, S., Clark, J. S., and Schimel, D. S.: Ecological forecasting and data assimilation in a data-rich era, Ecol. Appl., 21, 1429-1442, https://doi.org/10.1890/09-1275.1, 2011.
- Luo, Y., Weng, E., Wu, X., Gao, C., Zhou, X., and Zhang, L.: Parameter identifiability, constraint, and equifinality in data assimilation with ecosystem models, Ecol. Appl., 19, 571-574, https://doi.org/10.1890/08-0561.1, 2009.
- Luo, Y., White, L. W., Canadell, J. G., DeLucia, E. H., Ellsworth, D. S., Finzi, A., Lichter, J., and Schlesinger, W. H.: Sustainability of terrestrial carbon sequestration: A case study in Duke Forest with inversion approach, Global Biogeochem. Cy., 17, 2002GB001923, <a href="https://doi.org/10.1029/2002GB001923">https://doi.org/10.1029/2002GB001923</a>, 2003.
- Ma, S., Jiang, J., Huang, Y., Shi, Z., Wilson, R. M., Ricciuto, D., Sebestyen, S. D., Hanson, P. J., and Luo, Y.: Data-Constrained Projections of Methane Fluxes in a Northern Minnesota Peatland in Response to Elevated CO2 and Warming, J. Geophys. Res.-Biogeo., 122, 2841-2861, <a href="https://doi.org/10.1002/2017JG003932">https://doi.org/10.1002/2017JG003932</a>, 2017.
- Ma, S., Jiang, L., Wilson, R. M., Chanton, J. P., Bridgham, S., Niu, S., Iversen, C. M., Malhotra, A., Jiang, J., Lu, X., Huang, Y., Keller, J., Xu, X., Ricciuto, D. M., Hanson, P. J., and Luo, Y.: Evaluating alternative ebullition models for predicting peatland

- methane emission and its pathways via data-model fusion, Biogeosciences, 19, 2245-2262, <a href="https://doi.org/10.5194/bg-19-2245-2022">https://doi.org/10.5194/bg-19-2245-2022</a>, 2022.
- Ma, Z., Zhao, C., Gong, J., Zhang, J., Li, Z., Sun, J., Liu, Y., Chen, J., and Jiang, Q.: Spinup characteristics with three types of initial fields and the restart effects on forecast accuracy in the GRAPES global forecast system, Geoscientific Model Development, 14, 205-221, https://doi.org/10.5194/gmd-14-205-2021, 2021.
- Mao, R., Zeng, D. H., Zhang, X. H., and Song, C. C.: Responses of plant nutrient resorption to phosphorus addition in freshwater marsh of Northeast China, Scientific Reports, 5, 8097, https://doi.org/10.1038/srep08097, 2015.
- Marklein, A. R. and Houlton, B. Z.: Nitrogen inputs accelerate phosphorus cycling rates across a wide variety of terrestrial ecosystems, New Phytol., 193, 696-704, https://doi.org/10.1111/j.1469-8137.2011.03967.x, 2012.
- McGill, W. B., and Cole, C. V.: Comparative aspects of cycling of organic C, N, S and P through soil organic matter, Geoderma, 26, 267-286, <a href="https://doi.org/10.1016/0016-1307">https://doi.org/10.1016/0016-1307</a> 7061(81)90024-0, 1981.
- McGroddy, M. E., Daufresne, T., and Hedin, L. O.: Scaling of C:N:P stoichiometry in forests worldwide: Implications of terrestrial Redfield-type ratios, Ecology, 85, 2390-2401, <a href="https://doi.org/10.1890/03-0351">https://doi.org/10.1890/03-0351</a>, 2004.
- Medlyn, B. E., De Kauwe, M. G., Zaehle, S., Walker, A. P., Duursma, R. A., Luus, K., Mishurov, M., Pak, B., Smith, B., Wang, Y.-P., Yang, X., Crous, K. Y., Drake, J. E., Gimeno, T. E., Macdonald, C. A., Norby, R. J., Power, S. A., Tjoelker, M. G., and Ellsworth, D. S.: Using models to guide field experiments: a priori predictions for the CO2 response of a nutrient- and water-limited native Eucalypt woodland, Glob. Change Biol., 22, 2834-2851, https://doi.org/10.1111/gcb.13268, 2016.
- Meyerholt, J. and Zaehle, S.: The role of stoichiometric flexibility in modelling forest ecosystem responses to nitrogen fertilization, New Phytol., 208, 1042–1055, https://doi.org/10.1111/nph.13547, 2015.
- Nakhavali, M. A., Mercado, L. M., Hartley, I. P., Sitch, S., Cunha, F. V., Di Ponzio, R., and Camargo, J. L.: Representation of the phosphorus cycle in the Joint UK Land Environment Simulator (vn5.5\_JULES-CNP), Geosci. Model Dev., 15, 5241-5269, <a href="https://doi.org/10.5194/gmd-15-5241-2022">https://doi.org/10.5194/gmd-15-5241-2022</a>, 2022.
- Olander, L. P., and Vitousek, P. M.: Short-term controls over inorganic phosphorus during soil and ecosystem development, Soil Biol. Biochem., 37, 651-659, https://doi.org/10.1016/j.soilbio.2004.08.022, 2005.
- Pan, Y., Birdsey, R. A., Phillips, O. L., Houghton, R. A., Fang, J., Kauppi, P. E., Keith, H., Kurz, W. A., Ito, A., Lewis, S. L., Nabuurs, G.-J., Shvidenko, A., Hashimoto, S., Lerink, B., Schepaschenko, D., Castanho, A., and Murdiyarso, D.: The enduring world forest carbon sink, Nature, 631, 563-569, <a href="https://doi.org/10.1038/s41586-024-07602-x">https://doi.org/10.1038/s41586-024-07602-x</a>, 024.
- Parton, W. J., Hartman, M., Ojima, D., and Schimel, D.: DAYCENT and its land surface submodel: Description and testing, Global Planet. Change, 19, 35-48, <a href="https://doi.org/10.1016/S0921-8181(98)00040-X">https://doi.org/10.1016/S0921-8181(98)00040-X</a>, 1998.
- Pellitier, P. T., and Jackson, R. B.: Microbes modify soil nutrient availability and mediate plant responses to elevated CO2, Plant Soil, 483, 659-666, https://doi.org/10.1007/s11104-022-05807-5, 2023.

- Peng, Y., Peng, Z., Zeng, X., and Houx, J.: Effects of nitrogen-phosphorus imbalance on plant biomass production: A global perspective, Plant Soil, 436, 1-8, https://doi.org/10.1007/s11104-018-03927-5, 2019.
- Peñuelas, J., Poulter, B., Sardans, J., Ciais, P., van der Velde, M., Bopp, L., Boucher, O., 1342 Godderis, Y., Hinsinger, P., Llusia, J., Nardin, E., Vicca, S., Obersteiner, M., and 1343 Janssens, I. A.: Human-induced nitrogen-phosphorus imbalances alter natural and 1344 managed ecosystems across the globe, Nat. Commun.. 1345 https://doi.org/10.1038/ncomms3934, 2013.
- Rastetter, E. B.: Modeling coupled biogeochemical cycles, Front. Ecol. Environ., 9, 68-73, <a href="https://doi.org/10.1890/090223">https://doi.org/10.1890/090223</a>, 2011.
- Reed, S. C., Cleveland, C. C., and Townsend, A. R.: Functional ecology of free-living nitrogen fixation: A contemporary perspective, Annu. Rev. Ecol. Evol. Syst., 42, 489-512, <a href="https://doi.org/10.1146/annurev-ecolsys-102710-145034">https://doi.org/10.1146/annurev-ecolsys-102710-145034</a>, 2011.
- Reed, S. C., Townsend, A. R., Davidson, E. A., and Cleveland, C. C.: Stoichiometric patterns in foliar nutrient resorption across multiple scales, New Phytologist, 196, 173-180, <a href="https://doi.org/10.1111/j.1469-8137.2012.04249.x">https://doi.org/10.1111/j.1469-8137.2012.04249.x</a>, 2012.
- Requena Suarez, D., Rozendaal, D. M. A., De Sy, V., Phillips, O. L., Alvarez-Dávila, E., Anderson-Teixeira, K., Araujo-Murakami, A., Arroyo, L., Baker, T. R., and Bongers, F.: Estimating aboveground net biomass change for tropical and subtropical forests: Refinement of IPCC default rates using forest plot data, Glob. Change Biol., 25, 3609-3624, https://doi.org/10.1111/gcb.14767, 2019.
- Schachtman, D. P., Reid, R. J., and Ayling, S. M.: Phosphorus uptake by plants: from soil to cell, Plant Physiol., 116, 447-453, <a href="https://doi.org/10.1104/pp.116.2.447">https://doi.org/10.1104/pp.116.2.447</a>, 1998.
- Schleuss, P. M., Widdig, M., Heintz-Buschart, A., Kirkman, K., and Spohn, M.: Interactions of nitrogen and phosphorus cycling promote P acquisition and explain synergistic plant-growth responses, Ecology, 101, e03003, https://doi.org/10.1002/ecy.3003, 2020.
- Schubert, S., Steffens, D. and Ashraf, I.: Is occluded phosphate plant-available? J. Plant Nutr. Soil Sci., 183, 338-344, <a href="https://doi.org/10.1002/jpln.201900402">https://doi.org/10.1002/jpln.201900402</a>, 2020.
- See, C. R., Yanai, R. D., Fisk, M. C., Vadeboncoeur, M. A., Quintero, B. A., and Fahey, T. J.: Soil nitrogen affects phosphorus recycling: foliar resorption and plant-soil feedbacks in a northern hardwood forest, Ecology, 96, 2488-2498, https://doi.org/10.1890/14-1426.1, 2015.
- Shi, Z., Yang, Y., Zhou, X., Weng, E., Finzi, A. C., and Luo, Y.: Inverse analysis of coupled carbon-nitrogen cycles against multiple datasets at ambient and elevated CO2, J. Plant Ecol., 9, 285-295, https://doi.org/10.1093/jpe/rtv059, 2016.
- Sierra, C. A., Malghani, S., and Müller, M.: Model structure and parameter identification of soil organic matter models, Soil Biol. Biochem., 90, 197-203, https://doi.org/10.1016/j.soilbio.2015.07.018, 2015.
- Sitch, S., Smith, B., Prentice, I. C., Arneth, A., Bondeau, A., Cramer, W., Kaplan, J. O., Levis, S., Lucht, W., Sykes, M. T., Thonicke, K., and Venevsky, S.: Evaluation of ecosystem dynamics, plant geography and terrestrial carbon cycling in the LPJ dynamic global vegetation model, Global Change Biol., 9, 161-185,
- https://doi.org/10.1046/j.1365-2486.2003.00569.x, 2003.

- Smith, B., Prentice, I.C. and Sykes, M.T., Representation of vegetation dynamics in the modelling of terrestrial ecosystems: comparing two contrasting approaches within European climate space, Global Ecology and Biogeography, 10, 621-637. https://doi.org/10.1046/j.1466-822X.2001.t01-1-00256.x, 2001
- Smith, B., Wårlind, D., Arneth, A., Hickler, T., Leadley, P., Siltberg, J., and Zaehle, S.: Implications of incorporating N cycling and N limitations on primary production in an individual-based dynamic vegetation model, Biogeosciences, 11, 2027-2054, <a href="https://doi.org/10.5194/bg-11-2027-2014">https://doi.org/10.5194/bg-11-2027-2014</a>, 2014.
- Song, Y. C., and Wang, X. R.: Vegetation and flora of Tiantong National Forest Park Zhejiang Province, Shanghai Scientific and Technical Document Publishing House, Shanghai, 1995.
- Sterner, R. W., and Elser, J. J.: Ecological stoichiometry: the biology of elements from molecules to the biosphere, Princeton University Press, Princeton, 2002.
- Sun, Y., Peng, S., Goll, D. S., Ciais, P., Guenet, B., Guimberteau, M., Hinsinger, P., 1396 Janssens, I. A., Peñuelas, J., Piao, S., Poulter, B., Violette, A., Yang, X., Yin, Y., and 1397 Zeng, H. Diagnosing phosphorus limitations in natural terrestrial ecosystems in 1398 carbon cvcle models. Earth's Future, 5(7), 730-749. 1399 https://doi.org/10.1002/2016EF000472, 2017.

1402

- Tang, M., Zhao, W., Xing, M., Zhao, J., Jiang, Z., You, J., Ni, B., Ni, Y., Liu, C., Li, J., and Chen, X.: Resource allocation strategies among vegetative growth, sexual reproduction, asexual reproduction and defense during growing season of *Aconitum kusnezoffii* Reichb., Plant J., 105, 957-977, https://doi.org/10.1111/tpj.15080, 2021.
- Taylor, B. N., and Menge, D. N. L.: Light, nitrogen supply, and neighboring plants dictate costs and benefits of nitrogen fixation for seedlings of a tropical nitrogen-fixing tree, New Phytol., 231, 1758-1769, <a href="https://doi.org/10.1111/nph.17508">https://doi.org/10.1111/nph.17508</a>, 2021.
- Thomas, R. Q., Brookshire, E. N. J., and Gerber, S.: Nitrogen limitation on land: how can it occur in Earth system models?, Global Change Biol., 21, 1777-1793, <a href="https://doi.org/10.1111/gcb.12813">https://doi.org/10.1111/gcb.12813</a>, 2015.
- Tian, D., Yan, Z.-B., and Fang, J.-Y.: Review on characteristics and main hypotheses of plant ecological stoichiometry, Chinese Journal of Plant Ecology, 45, 682-713, https://doi.org/10.17521/cjpe.2020.0331, 2021.
- Tian, H., Chen, G., Zhang, C., Melillo, J. M., and Hall, C. A. S.: Pattern and variation of C:N:P ratios in China's soils: a synthesis of observational data, Biogeochemistry, 98, 139-151, https://doi.org/10.1007/s10533-009-9382-0, 2010.
- Treseder, K., and Vitousek, P. M.: Effects of soil nutrient availability on investment in acquisition of N and P in Hawaii rain forests, Ecology, 82, 946-954, <a href="https://doi.org/10.1890/0012-9658(2001)082[0946:EOSNAO]2.0.CO;2">https://doi.org/10.1890/0012-9658(2001)082[0946:EOSNAO]2.0.CO;2</a>, 2001.
- Veneklaas, E. J., Lambers, H., Bragg, J., Finnegan, P. M., Lovelock, C. E., Plaxton, W. C., and Raven, J. A.: Opportunities for improving phosphorus-use efficiency in crop plants, New Phytol., 195, 306-320, <a href="https://doi.org/10.1111/j.1469-8137.2012.04190.x">https://doi.org/10.1111/j.1469-8137.2012.04190.x</a>, 2012.
- Vitousek, P. M., Menge, D. N. L., Reed, S. C., and Cleveland, C. C.: Biological nitrogen fixation: Rates, patterns and ecological controls in terrestrial ecosystems, Philos. T. R. Soc. B, 368, 20130119, https://doi.org/10.1098/rstb.2013.0119, 2013.

- Vitousek, P. M., Porder, S., Houlton, B. Z., and Chadwick, O. A.: Terrestrial phosphorus limitation: mechanisms, implications, and nitrogen–phosphorus interactions, Ecol. Appl., 20, 5-15, 2010.
- Walker, A. P., Beckerman, A. P., Gu, L., Kattge, J., Cernusak, L. A., Domingues, T. F., Scales, J. C., Wohlfahrt, G., Wullschleger, S. D., and Woodward, F. I.: The relationship of leaf photosynthetic traits V cmax and J max to leaf nitrogen, leaf phosphorus, and specific leaf area: A meta-analysis and modeling study, Ecol. Evol., 4, 3218-3235, https://doi.org/10.1002/ece3.1173, 2014.
- Walker, T. W., and Syers, J. K.: The fate of phosphorus during pedogenesis, Geoderma, 1435 15, 1-19, <a href="https://doi.org/10.1016/0016-7061(76)90066-5">https://doi.org/10.1016/0016-7061(76)90066-5</a>, 1976.
- Walton, C. R., Ewens, S., Coates, J. D., and Chen, K.: Phosphorus availability on the early
  Earth and the impacts of life, Nat. Geosci., 16, 399-409,
  https://doi.org/10.1038/s41561-023-01167-6, 2023.
- Wan, F. X.: FangxiuWan/TECO-CNP-Sv1.0: TECO-CNP Sv1.0 (Sv1.0). Zenodo. 1440 <a href="https://doi.org/10.5281/zenodo.15032706">https://doi.org/10.5281/zenodo.15032706</a>, 2025a.
- Wan, F. X.: FangxiuWan/ModelOutput\_TECO-CNP-Sv1.0: TECO-CNP Sv 1.0 Output Example (Data). Zenodo. <a href="https://doi.org/10.5281/zenodo.15033861">https://doi.org/10.5281/zenodo.15033861</a>, 2025b.
- Wan, F. X.: FangxiuWan/FIguresCode\_4\_TECO-CNP-Sv-1.0: v1.0 (v1.0). Zenodo. 1444 <a href="https://doi.org/10.5281/zenodo.15032690">https://doi.org/10.5281/zenodo.15032690</a>, 2025c.
- Wang, Q.: Adsorption-desorption characteristics of exogenous phosphorus in subtropical evergreen broad-leaved forest soil under extreme drought and the associated microbial mechanisms, Master's thesis, East China Normal University, Shanghai, China, 2022.
- Wang, S., Luo, Y., and Niu, S.: Reparameterization required after model structure changes from carbon only to carbon-nitrogen coupling, J. Adv. Model. Earth Syst., 14, e2021MS002798, https://doi.org/10.1029/2021MS002798, 2022.

- Wang, Q.: Adsorption-desorption characteristics of exogenous phosphorus in subtropical evergreen broad-leaved forest soil under extreme drought and the associated microbial mechanisms, Master's thesis, East China Normal University, Shanghai, China, 2022.
- Wang, X., Li, S., Zhu, B., Homyak, P. M., Chen, G., Yao, X., Wu, D., Yang, Z., Lyu, M., and Yang, Y.: Long-term nitrogen deposition inhibits soil priming effects by enhancing phosphorus limitation in a subtropical forest, Glob. Change Biol., 29, 4081-4093, https://doi.org/10.1111/gcb.16718, 2023.
- Wang, X., Li, Z., and Zhang, F.: An improved instantaneous gross primary productivity model considering the difference in contributions of sunlit and shaded leaves to canopy sun-induced chlorophyll fluorescence, Global and Planetary Change, 243, 104627, https://doi.org/10.1016/j.gloplacha.2024.104627, 2024.
- Wang, Y. P., and Leuning, R.: A two-leaf model for canopy conductance, photosynthesis and partitioning of available energy I: Model description and comparison with a multi-layered model, Agr. Forest Meteorol., 91, 89-111, <a href="https://doi.org/10.1016/S0168-1923(98)00061-6">https://doi.org/10.1016/S0168-1923(98)00061-6</a>, 1998.
- Wang, Y. P., Law, R. M., and Pak, B.: A global model of carbon, nitrogen and phosphorus cycles for the terrestrial biosphere, Biogeosciences, 7, 2261-2282, <a href="https://doi.org/10.5194/bg-7-2261-2010">https://doi.org/10.5194/bg-7-2261-2010</a>, 2010.
- Wang, Y., Huang, Y., Song, L., Yuan, J., Li, W., Zhu, Y., Chang, S. X., Luo, Y., Ciais, P.,
   Peñuelas, J., Wolf, J., Cade-Menun, B. J., Hu, S., Wang, L., Wang, D., Yuan, Z.,

- Wang, Y., Zhang, J., Tao, Y., and Zhu, C.: Reduced phosphorus availability in paddy soils under atmospheric CO2 enrichment, Nat. Geosci., 16, 162-168, https://doi.org/10.1038/s41561-022-01105-y, 2023.
- Wang, Z., Tian, H., Pan, S., Shi, H., Yang, J., Liang, N., Kalin, L., and Anderson, C.:
  Phosphorus limitation on CO2 fertilization effect in tropical forests informed by a
  coupled biogeochemical model, Forest Ecosyst., 11, 100210,
  <a href="https://doi.org/10.1016/j.fecs.2024.100210">https://doi.org/10.1016/j.fecs.2024.100210</a>, 2024.
- Wassen, M. J., de Boer, H. J., Fleischer, K., Rebel, K. T., and Dekker, S. C.: Vegetation-mediated feedback in water, carbon, nitrogen and phosphorus cycles, Landscape Ecol., 28, 599-614, <a href="https://doi.org/10.1007/s10980-012-9843-z">https://doi.org/10.1007/s10980-012-9843-z</a>, 2013.
- Wei, N., and Xia, J. Y.: Robust projections of increasing land carbon storage in boreal and temperate forests under future climate change scenarios, One Earth, 7, 88-99, https://doi.org/10.1016/j.oneear.2023.11.013, 2024.
- Wei, N., Xia, J., Wang, Y.-P., Zhang, X., Zhou, J., Bian, C., and Luo, Y., Nutrient limitations lead to a reduced magnitude of disequilibrium in the global terrestrial carbon cycle. Journal of Geophysical Research: Biogeosciences, 127, e2021JG006764. https://doi.org/10.1029/2021JG006764, 2022.
- Weihrauch, C., and Opp, C.: Ecologically relevant phosphorus pools in soils and their dynamics: The story so far. Geoderma, 325, 183–194. <a href="https://doi.org/10.1016/j.geoderma.2018.02.047">https://doi.org/10.1016/j.geoderma.2018.02.047</a>, 2018.
- Weng, E., and Luo, Y.: Soil hydrological properties regulate grassland ecosystem responses to multifactor global change: A modeling analysis, J. Geophys. Res., 113, G03003, <a href="https://doi.org/10.1029/2007JG000539">https://doi.org/10.1029/2007JG000539</a>, 2008.
- Weng, E., Farrior, C. E., Dybzinski, R., and Pacala, S. W.: Predicting vegetation type through physiological and environmental interactions with leaf traits: evergreen and deciduous forests in an earth system modeling framework, Global Change Biol., 23, 2482-2498, <a href="https://doi.org/10.1111/gcb.13542">https://doi.org/10.1111/gcb.13542</a>, 2017.
- Wieder, W. R., Cleveland, C. C., Smith, W. K., and Todd-Brown, K.: Future productivity and carbon storage limited by terrestrial nutrient availability, Nat. Geosci., 8, 441-444, <a href="https://doi.org/10.1038/ngeo2413">https://doi.org/10.1038/ngeo2413</a>, 2015.
- Wiltshire, A. J., Burke, E. J., Chadburn, S. E., Jones, C. D., Cox, P. M., Davies-Barnard, T., Friedlingstein, P., Harper, A. B., Liddicoat, S., Sitch, S., and Zaehle, S.: JULES-CN: A coupled terrestrial carbon-nitrogen scheme (JULES vn5.1), Geosci. Model Dev., 14, 2161-2186, <a href="https://doi.org/10.5194/gmd-14-2161-2021">https://doi.org/10.5194/gmd-14-2161-2021</a>, 2021.
- Wu, X., Liang, Y., Zhao, W., and Pan, F.: Root and mycorrhizal nutrient acquisition strategies in the succession of subtropical forests under N and P limitation. BMC Plant Biology, 25(1), 8, <a href="https://doi.org/10.1186/s12870-024-06016-1">https://doi.org/10.1186/s12870-024-06016-1</a>, 2025.
- Xia, J., Chen, Y., Yuan, W., and Wang, Y.-P.: The effects of multiple environmental factors on global carbon allocation, Ecol. Process., 12, 60, https://doi.org/10.1186/s13717-023-00477-2, 2023.
- Xia, J., Wan, S.: Global response patterns of terrestrial plant species to nitrogen addition.

  New Phytol., 179, 428-439, <a href="https://doi.org/10.1111/j.1469-8137.2008.02488.x">https://doi.org/10.1111/j.1469-8137.2008.02488.x</a>, 2008.
- Xu, T., White, L., Hui, D., and Luo, Y.: Probabilistic inversion of a terrestrial ecosystem model: Analysis of uncertainty in parameter estimation and model prediction, Global Biogeochem. Cy., 20, 2005GB002468, https://doi.org/10.1029/2005GB002468, 2006.

- Xu, X., Xia, J., Zhou, X., and Yan, L.: Experimental evidence for weakened tree nutrient use and resorption efficiencies under severe drought in a subtropical monsoon forest, J. Plant Ecol., 13, 461-469, https://doi.org/10.1093/jpe/rtaa053, 2020.
- Yan, B., Ji, Z., Fan, B., Wang, X., He, G., Shi, L., and Liu, G.: Plants adapted to nutrient limitation allocate less biomass into stems in an arid-hot grassland, New Phytol., 211, 1232-1240, https://doi.org/10.1111/nph.13970, 2016.
- Yan, H., Wang, S. Q., Yu, K. L., Wang, B., Yu, Q., Bohrer, G., and Shugart, H. H.: A novel diffuse fraction-based two-leaf light use efficiency model: An application quantifying photosynthetic seasonality across 20 AmeriFlux flux tower sites, Journal of Advances in Modeling Earth Systems, 9, 2317-2332, <a href="https://doi.org/10.1002/2016MS000886">https://doi.org/10.1002/2016MS000886</a>, 2017.
- Yang, Q., Yang, H., Zheng, Z., Liu, H., Yao, F., Jiang, S., and Wang, X.: A dataset of species composition and biomass in different successional stages of Tiantong typical evergreen broad-leaved forests (2008–2017), Science Data Bank, https://doi.org/10.57760/sciencedb.j00001.00695, 2022.
- Yang, X., Post, W. M., Thornton, P. E., and Jain, A.: The distribution of soil phosphorus for global biogeochemical modeling, Biogeosciences, 10, 2525-2537, https://doi.org/10.5194/bg-10-2525-2013, 2013.
- Yang, X., Thornton, P. E., Ricciuto, D. M., and Post, W. M.: The role of phosphorus dynamics in tropical forests—a modeling study using CLM-CNP, Biogeosciences, 11, 1636 1667-1681, <a href="https://doi.org/10.5194/bg-11-1667-2014">https://doi.org/10.5194/bg-11-1667-2014</a>, 2014.
- Yu, G., Chen, Z., Piao, S., Peng, C., Ciais, P., Wang, Q., Li, X., and Zhu, X.: High carbon dioxide uptake by subtropical forest ecosystems in the East Asian monsoon region, Proc. Natl. Acad. Sci. USA, 111, 4910-4915, https://doi.org/10.1073/pnas.1317065111, 2014.
- Zaehle, S., Medlyn, B. E., De Kauwe, M. G., Walker, A. P., Dietze, M. C., Hickler, T.,
  Luo, Y., Wang, Y., El-Masri, B., Thornton, P., Jain, A., Wang, S., Warlind, D., Weng,
  E., Parton, W., Iversen, C. M., Gallet-Budynek, A., McCarthy, H., Finzi, A., and
  Norby, R. J.: Evaluation of 11 terrestrial carbon-nitrogen cycle models against
  observations from two temperate Free-Air CO2 Enrichment studies, New Phytol., 202,
  803-822, https://doi.org/10.1111/nph.12697, 2014.
- Zavišić, A., and Polle, A.: Dynamics of phosphorus nutrition, allocation and growth of young beech (*Fagus sylvatica L*.) trees in P-rich and P-poor forest soil, Tree Physiol., 38, 37-51, https://doi.org/10.1093/treephys/tpx146, 2018.
- Zeng, F. R., Shi, J. Y., Yan, E. R., Zhang, R. L., and Wang, X. H.: Temporal and spatial patterns of fine root mass along a secondary succession of evergreen broad-leaved forest in Tiantong, Journal of East China Normal University (Natural Science), 2008, 56-62, 2008.
- Zheng, Z. M., Yang, H. B., Dong, S., Yao, F. F., Yang, Q. S., Wang, X. H., Yan, E. R., and
   Jiang, S.: Monthly and annual litterfall production dataset across different
   successional stages of evergreen broad-leaved forest in Tiantong from 2008 to 2018,
   China Sci. Data, 8, 259-267, 2023.
- Zhou, G.: Responses of soil respiration to simulated drought and nitrogen addition in a subtropical evergreen broad-leaved forest, Ph.D. thesis, East China Normal University, Shanghai, China, 2020.

- Zhou, J., Chen, S., Yan, L., Wang, J., Jiang, M., Liang, J., Zhang, X., and Xia, J.: A
  Comparison of Linear Conventional and Nonlinear Microbial Models for Simulating
  Pulse Dynamics of Soil Heterotrophic Respiration in a Semi-Arid Grassland, J.
  Geophys. Res.-Biogeo., 126, e2020JG006120,
  https://doi.org/10.1029/2020JG006120, 2021.
- Zhu, H., Zhao, J., and Gong, L.: The morphological and chemical properties of fine roots respond to nitrogen addition in a temperate Schrenk's spruce (*Picea schrenkiana*) forest, Sci. Rep.-UK, 11, 3839, <a href="https://doi.org/10.1038/s41598-021-83151-x">https://doi.org/10.1038/s41598-021-83151-x</a>, 2021.
- Zhu, J., Wang, Q., He, N., Smith, M. D., Elser, J. J., Du, J., Yuan, G., Yu, G., and Yu, Q.:
  Imbalanced atmospheric nitrogen and phosphorus depositions in China: Implications for nutrient limitation, J. Geophys. Res.-Biogeo., 121, 1605-1616, <a href="https://doi.org/10.1002/2016JG003393">https://doi.org/10.1002/2016JG003393</a>, 2016.

1574

- Zhu, Q., Riley, W. J., Tang, J., and Koven, C. D.: Multiple soil nutrient competition between plants, microbes, and mineral surfaces: Model development, parameterization, and example applications in several tropical forests, Biogeosciences, 13, 341-363, <a href="https://doi.org/10.5194/bg-13-341-2016">https://doi.org/10.5194/bg-13-341-2016</a>, 2016.
- Zhu, Q., Riley, W. J., Tang, J., Collier, N., Hoffman, F. M., Yang, X., and Bisht, G.: Representing Nitrogen, Phosphorus, and Carbon Interactions in the E3SM Land Model: Development and Global Benchmarking, J. Adv. Model. Earth Syst., 11, 2238-2258, https://doi.org/10.1029/2018MS001571, 2019.
- Zust, T., and Agrawal, A. A.: Trade-Offs Between Plant Growth and Defense Against Insect Herbivory: An Emerging Mechanistic Synthesis, Annu. Rev. Plant Biol., 68, 513-534, <a href="https://doi.org/10.1146/annurev-arplant-042916-040856">https://doi.org/10.1146/annurev-arplant-042916-040856</a>, 2017.
- Zust, T., Rasmann, S., and Agrawal, A. A.: Growth-defense tradeoffs for two major antiherbivore traits of the common milkweed Asclepias syriaca, Oikos, 124, 1404-1415, <a href="https://doi.org/10.1111/oik.02075">https://doi.org/10.1111/oik.02075</a>, 2015.