# Peer review of "(untitled)"

_EGUsphere, 2025_

## Author Comment (AC1)

**Responses to comments from reviewer #1**
(Manuscript number: egusphere-2025-1243)

We sincerely thank Reviewer 1 for the thoughtful and constructive comments. In response, we have revised the manuscript to clarify parameterization strategies, explain the integration of process-based modeling with data assimilation, and refine the definitions and representations of soil P pools. Reviewer comments are shown in blue italic, followed by our detailed responses. We hope these revisions address all concerns satisfactorily.

**Comment 1A:** *This is a nicely written and well-executed work implementing phosphorus cycle and data assimilation framework into a process-based ecosystem model (TECO). The novelty of this work lies in the model development and its coupling with data assimilation. By comparing CNP model against observations and their respective C-only and CN coupled models, the authors show a superior performance of the newly developed model. Overall, I enjoy reading the manuscript, and support its publication. There are several occasions where I think some justifications/modifications would further improve the quality of the manuscript. Below I list my main questions/concerns.*

**Response:** We sincerely appreciate your positive assessment of our manuscript and the constructive feedback provided. We have carefully considered all the questions and concerns raised by the reviewer and have made substantial revisions to address these points.

**Comment 2A:** *There are certain processes where N and P interact. For example, some models consider N to have an effect on soil P biochemical mineralization rate (e.g. ORCHIDEE, ELM, etc.). It does not seem that the authors have adopted these NP interacting processes in their model. Furthermore, others consider N and P to have joint effect on C processes, such as photosynthesis. In this work, it seems that nutrient effect on photosynthesis is realized via downregulation of leaf surface area. There have been some empirical relationships derived on how N and P affects photosynthetic traits (e.g. Vcmax, Jmax; Ellsworth et al., 2022; Walker et al, 2014), and these relationships have been incorporated into models. What is the authors' consideration on not following these conventional approaches?*

**Response:** Thank you for raising this important point regarding nitrogen-phosphorus (N-P) interactions and nutrient regulation of photosynthesis. We acknowledge that N-P interactions are central to ecosystem function and have explicitly incorporated two key mechanisms into the TECO-CNP model: (1) nitrogen-limited phosphorus uptake regulated by a nutrient balance scalar (Eqs. 26-27), and (2) a cost-benefit mechanism for phosphatase production (Eq. 43), whereby plants optimize P acquisition strategy based on nitrogen cost. We have now clarified these processes in the revised manuscript (Section 2.2.2, L364; Section 2.2.3, L470-480) and added a general summary in Section 2.2.2 (L278-281) to improve readability.

We agree with you that TECO-CNP currently offers a partial representation of N-P interactions, reflecting broader challenges in coupled CNP modeling (Achat et al., 2016). Although recent studies have advanced our understanding of N-P coupling mechanisms (Luo et al., 2022; Krouk & Kiba, 2020; Hu & Chu, 2019), their integration into models remains limited by both incomplete mechanistic knowledge and the scarcity of empirical data for parameterization. For example, complex relationships such as the coordination between biological nitrogen fixation and phosphatase production (Batterman et al., 2018) and the reciprocal effects of nitrogen and phosphorus availability on nutrient resorption dynamics (See et al., 2015; Li et al., 2019) are still

poorly understood. Thus, we view advancing the representation of N-P interactions as a key priority for future model development. We will continue to incorporate these processes as our mechanistic understanding deepens and more comprehensive datasets become available. We have added several sentences in the revised version to discuss this point in the revised version (L829-833, L840-852).

Regarding the empirical relationships developed by Ellsworth et al. (2022) and Walker et al. (2014) linking N and P to photosynthetic parameters ($V_{cmax}$, $J_{max}$), we agree that these empirical relationships would be an advancement for CNP models, and we carefully considered incorporating these relationships after receiving your comments. However, we ultimately decided to adopt this approach in our future versions due to two reasons. First, we tested these relationships, which are mainly derived from large-scale and cross-biome analyses, at our site. As shown in the Table S5, adding these empirical relationships resulted in significant overestimation of photosynthetic parameters. Second, our site-specific dataset was insufficient to robustly generate similar relationships between photosynthetic parameters and leaf nutrient content, and applying unvalidated relationships could introduce additional uncertainty into model simulations. Therefore, we implemented nutrient limitation on photosynthesis indirectly through plant growth regulation (Eq. 1, L173-189), which we believe is more appropriate for our site-scale application given the available data constraints. We refine our discussion about the nutrient-regulated photosynthesis capacity in Section 3.1.1 at L747-757 as follows:

"The observed reduction in LAI represents a decrease in photosynthetic capacity achieved through nutrient limitation of plant growth, which reduces the photosynthetic leaf area rather than directly affecting leaf-level photosynthetic physiological parameters. The relationships between leaf nitrogen and phosphorus concentrations and photosynthetic traits (e.g., Vcmax, Jmax) are well established (Walker et al., 2014; Ellsworth et al., 2022) and have been incorporated into some land surface models (e.g., JULES-CNP). However, these large-scale emergent relationships significantly overestimated photosynthetic parameters at our study site (Table S5). At the same time, our site-specific dataset was insufficient to derive robust empirical relationships between nutrient concentrations and photosynthetic capacity. Future studies with more comprehensive site-level measurements could enhance this aspect of the model to represent nutrient-carbon interactions better"

**Comment 3A:** *Solution P is part of labile P, and some work suggests the need to explicitly model solution P in addition to labile P (e.g. Reed et al., 2015). In this work, how does the author consider this suggestion and what is the rationale for only simulating labile P?*

**Response:** Thank you for this important question regarding the rationale for inorganic P (Pi) pool representation in TECO-CNP. Also, many thanks for recommending the great paper, i.e., Reed et al., (2015). We agree with you and Reed et al. (2015) that more detailed representation of Pi pools could enhance model performance. In this revised version, we cited Reed et al., (2015) and clearly explained why only labile P was simulated at our site. Here are two brief reasons:

First, Pi extracted by resin bags or strips (resin-Pi) and extracted by $NaHCO_3$ (Hou et al., 2018) are typically considered as phosphorus readily available to plants and microorganisms, and they exhibit highly correlated dynamics with similar bioavailability characteristics (Hou et al., 2019). Employing a single P pool to represent the rapidly accessible phosphorus for plants and microorganisms in soil is common practice in ecosystem modeling studies (Wang et al., 2010, Wang et al., 2018; Hou et al., 2019; Nakhavali et al., 2022; Knox et al., 2024). However, as pointed

out by Reed et al. (2015), solution P need to explicitly modeled in addition to labile P. We have added several sentences to discuss this topic in this version (L510-520).

Second, most field datasets provide total labile P rather than separated fractions, as in our study (see Section 2.3.2 in the revised manuscript, L626-631). Therefore, implementing more complex processes without sufficient supporting data could increase model uncertainty without significantly improving model performance. At this stage, utilizing a single labile P pool rather than including a separate solution P pool represents a robust and conservative approach. In future developments, more detailed model structures will be considered as relevant research data become available.

Furthermore, to clarify the definitions of these Pi pools, we have revised the description at L286-290 as follows:

"Labile P represents readily bioavailable inorganic phosphate for rapid biological uptake. Sorbed P is weakly bound to soil surfaces in dynamic equilibrium with labile P. Through petrochemical processes, sorbed P transforms into secondary mineral P, which eventually becomes occluded P with minimal bioavailability"

**Comment 4A:** *It seems that the data assimilation framework was only applied to the CNP model, and then the authors reported CN and C models to overestimate observations. I find this logic to be a bit problematic. Without data assimilation, does CNP model still achieve better match with observations? Alternatively, how does C-only model coupled with data assimilation perform relative to observations? If it can achieve similar performance as compared to CNP model, what benefits of having a CNP model?*

**Response:** Thank you for raising these important questions regarding model comparison and the application of data assimilation. Please find our point-by-point replies as below:

(1) *Does the CNP model achieve better agreement with observations even without data assimilation*? Yes, as shown in Section 3.1 (Fig. 5, Table 5), the CNP model outperformed both the CN and C-only models in simulating observed carbon, nitrogen, and phosphorus pools and fluxes without any parameter optimization. This supports our initial hypothesis that including explicit N and P cycles improves model fidelity at the site level.

(2) *How does the C-only model perform when data assimilation is applied?* We conducted a supplementary analysis applying data assimilation to the C-only model and found that its performance in simulating carbon fluxes (e.g., GPP) was similar to that of the CNP model (Fig. RA1; Fig. S1 in revised supplementary materials). This result illustrates the well-known issue of equifinality, where structurally different models can achieve comparable predictive accuracy by compensating with different parameter sets.

[Figure]

**Figure S1.** Diurnal GPP patterns across months simulated by three model configurations (C, CN, CNP) versus observations. RMSE values for each model (green: C, blue: CN, red: CNP) are shown in the upper right of each panel. Black lines with grey shading represent observations (± 1 SD); colored lines and shading represent model means ± SD.

(3) *What is the added value of the CNP model, if performance appears similar?* The critical distinction lies in mechanistic realism. Although the optimized C-only model fits the observations, it does so by using ecologically implausible parameter values. For instance, the constrained specific leaf area (SLA) in the C-only model deviates substantially from the observed community-level distribution, unlike the CNP model (Fig. RA2; Fig. S2 in revised supplementary materials). This limits the ecological interpretability and transferability of the optimized C-only model, especially under changing environmental conditions where nutrient dynamics play a key role.

[Figure]

**Figure S2**. Posterior distribution of constrained specific leaf area (SLA) and observed community-level SLA distribution. Green, blue, red, and gray represent C, CN, CNP models and observations, respectively. Vertical lines represent distribution means. D, Kolmogorov-Smirnov statistic.

Therefore, the CNP model not only improves baseline simulation accuracy but also preserves mechanistic fidelity, enabling more reliable predictions under future climate and nutrient limitation scenarios. We have added further discussion on this point in Section 3.2 of the revised manuscript and detailed parameter comparison results in the Supplementary Materials (Text S1, Figs. S1–S2).

**Reference**

Achat, D. L., Augusto, L., Gallet-Budynek, A., & Loustau, D. (2016). Future challenges in coupled C–N–P cycle models for terrestrial ecosystems under global change: a review. Biogeochemistry, 131, 173–202. https://doi.org/10.1007/s10533-016-0274-9.

Batterman, S. A., Hall, J. S., Turner, B. L., Hedin, L. O., LaHaela Walter, J. K., Sheldon, P., & Van Breugel, M. (2018). Phosphatase activity and nitrogen fixation reflect species differences, not nutrient trading or nutrient balance, across tropical rainforest trees. Ecology Letters, 21(10), 1486–1495. https://doi.org/10.1111/ele.13129.

Hou, E., Lu, X., Jiang, L., Wen, D., & Luo, Y. (2019). Quantifying Soil Phosphorus Dynamics: A Data Assimilation Approach. Journal of Geophysical Research: Biogeosciences, 124(7), 2159–2173. https://doi.org/10.1029/2018JG004903.

Hou, E., Tan, X., Heenan, M., & Wen, D. (2018). A global dataset of plant available and unavailable phosphorus in natural soils derived by Hedley method. Scientific Data, 5(1). https://doi.org/10.1038/sdata.2018.166.

Hu, B., & Chu, C. (2020). Nitrogen–phosphorus interplay: Old story with molecular tale. New Phytologist, 225(4), 1455–1460. https://doi.org/10.1111/nph.16102.

Knox, R. G., Koven, C. D., Riley, W. J., Walker, A. P., Wright, S. J., & Holm, J. A. (2024). Nutrient dynamics in a coupled terrestrial biosphere and land model (ELM-FATES-CNP). Journal of Advances in Modeling Earth Systems, 16, e2023MS003689. https://doi.org/10.1029/2023MS003689.

Krouk, G., & Kiba, T. (2020). Nitrogen and Phosphorus interactions in plants: From agronomic to physiological and molecular insights. Current Opinion in Plant Biology, 57, 104–109. https://doi.org/10.1016/j.pbi.2020.07.002.

Li, L., Liu, B., Gao, X. et al. Nitrogen and phosphorus addition differentially affect plant ecological stoichiometry in desert grassland. Sci Rep 9, 18673 (2019). https://doi.org/10.1038/s41598-019-55275-8.

Luo, M., Moorhead, D. L., Ochoa-Hueso, R., Mueller, C. W., Ying, S. C., & Chen, J. (2022). Nitrogen loading enhances phosphorus limitation in terrestrial ecosystems with implications for soil carbon cycling. Functional Ecology, 36(11), 2845–2858. https://doi.org/10.1111/1365-2435.14178.

Nakhavali, M. A., Mercado, L. M., Hartley, I. P., Sitch, S., Cunha, F. V., Di Ponzio, R., & Camargo, J. L. (2022). Representation of the phosphorus cycle in the Joint UK Land Environment Simulator (vn5.5_JULES-CNP). Geoscientific Model Development, 15, 5241–5269. https://doi.org/10.5194/gmd-15-5241-2022.

See CR, Yanai RD, Fisk MC, Vadeboncoeur MA, Quintero BA, Fahey TJ (2015) Soil nitrogen affects phosphorus recycling: foliar resorption and plant–soil feedbacks in a northern hardwood forest. Ecology 96:2488–2498.

Wang, Q. (2022). Adsorption-desorption characteristics of exogenous phosphorus in subtropical evergreen broad-leaved forest soil under extreme drought and the associated microbial mechanisms. Master's thesis, East China Normal University, Shanghai, China.

Wang, Y. P., Law, R. M., & Pak, B. (2010). A global model of carbon, nitrogen and phosphorus cycles for the terrestrial biosphere. Biogeosciences, 7, 2261–2282. https://doi.org/10.5194/bg-7-2261-2010.

Wang, Y., Ciais, P., Goll, D., Huang, Y., Luo, Y., Wang, Y.-P., Bloom, A. A., Broquet, G., Hartmann, J., Peng, S., Penuelas, J., Piao, S., Sardans, J., Stocker, B. D., Wang, R., Zaehle, S., & Zechmeister-Boltenstern, S. (2018). GOLUM-CNP v1.0: A data-driven modeling of carbon, nitrogen and phosphorus cycles in major terrestrial biomes. Geoscientific Model Development, 11(9), 3903–3928. https://doi.org/10.5194/gmd-11-3903-2018.

---

## Author Comment (AC2)

**Responses to comments from reviewer #2**
(Manuscript number: egusphere-2025-1243)

We sincerely thank Reviewer 2 for the thoughtful and constructive comments. In response, we have revised the manuscript to clarify parameterization strategies, explain the integration of process-based modeling with data assimilation, and refine the definitions and representations of soil P pools. Reviewer comments are shown in blue italic, followed by our detailed responses. We hope these revisions address all concerns satisfactorily.

**Comment 1B:** *This study developed a coupled carbon-nitrogen-phosphorus model, TECO-CNP Sv1.0, based on the Terrestrial ECOsystem (TECO) model. The developed model was used to simulate C, N, and P pools and fluxes in a phosphorus-limited subtropical forest site in East China. In addition, a parameter optimization algorithm was also incorporated into the model framework to improve the model's performance. Overall, the manuscript provides detailed information on the model structure, parameters, and performance. However, I still have some questions on the soil pool structure and calibration processes of the model.*

**Response:** Thank you for your constructive review and valuable comments on our TECO-CNP model development. We are committed to addressing your concerns and providing clarifications that will strengthen our manuscript.

**Comment 2B:** *Four inorganic P pools, including labile P, sorbed P, secondary P, and occluded P, are set in TECO-CNP. This structure is different from other CNP models. For example, the labile P pool in ORCHIDEE-CNP includes both dissolved and sorbed P. In CLM-CNP, inorganic P pools include labile P (including solution P), secondary P, and occluded P. In a global P dataset developed by He et al. (2023) Biogeosciences, the soil inorganic P is divided into labile P, moderate P, and occluded P. Can you explain the differences in inorganic P pool structure among these models? I am confused about the definition of labile P pool. In addition, how did you initialize these inorganic P pools ?*

**Response:** We have added the detailed description of labile P pool in revised version (L286-290, L626-631). The key difference among models in the soil inorganic P pools is how they define and term the most available inorganic P pools (Table RB1). In some model studies, labile P serves as the directly plant-available pool (e.g., TECO-CNP, CASA-CNP, JSBACH-CNP) and maintains dynamic equilibrium with the sorbed pool over short timescales. In contrast, other studies tend to use terms like "solution" or "dissolved" to represent the solution-phase P (e.g., CLM-CNP, ORCHIDEE-CNP, E3SM-CNP), while also setting up another pool that equilibrates with it. The model structure is comparable to experimental approaches. Experimentally, "labile P" represents the inorganic P extracted by resin and $NaHCO_3$, and "secondary mineral P" represents NaOH-extracted inorganic P, which some studies term "moderately available P" (He et al., 2023).

We initialized inorganic P pools using site measurements. Labile P was determined from 0-10 cm soil samples collected in 2023 from a nearby forest stand of similar stand age (~200 yr) dominated by the same species as the Tiantong forest dynamic plot. Secondary P and occluded P values were obtained from available literature data for the same study site (Table S3). The initialization method is refined in the revised version (L625-634).

**Table RB1**. Comparison of inorganic phosphorus (P) pools in ORCHIDEE-CNP, CLM-CNP, and TECO-CNP. Bold text highlights the main differences among models.

| ORCHIDEE-CNP | CLM-CNP | TECO-CNP | General description |
|---|---|---|---|
| Dissolved labile P | Solution P | Labile P | Most readily available P and only source for plants uptake; can be adsorbed or lost by leaching |
| Sorbed labile P | Labile P | Sorbed P | P reversibly adsorbed onto soil particles; maintains equilibrium with the most readily available P pool |
| Secondary mineral P | Secondary mineral P | Secondary mineral P | Moderately stable P; can be slowly dissolved to enter labile pool or become occluded |
| Occluded P | Occluded P | Occluded P | Most stable P form; encapsulated by Fe/Al oxides; extremely slow release over geological timescales |

**Comment 3B:** *What is the advantage of TECO-CNP compared with other CNP models?*

**Response:** Thank you for this insightful question. In this revised version, we have made it clearer that TECO-CNP has three major advantages compared with other CNP models. First, TECO-CNP tightly couples vegetation carbon processes with soil nutrient cycling. It simulates dynamic plant growth responses to both soil nutrient availability and internal physiological traits by modifying growth rates (Eq. 1) and allocation patterns (Eqs. 9-11). This allows for a more mechanistic representation of nutrient-limited growth dynamics.

Second, unlike many current CNP models that omit non-structural carbohydrate (NSC) pools (e.g., JULES-CNP, CABLE, ELMv1-ECA; Nakhavali et al., 2022; Haverd et al., 2018; Zhu et al., 2019), TECO-CNP explicitly represents NSC dynamics. This enables a more realistic representation of how plants adjust allocation between growth and storage under nutrient stress (L103-106; Hartmann et al., 2020; Merganičová et al., 2019).

The third advantage of TECO-CNP compared to other CNP models lies in its capacity to integrate in situ observations with process-level forecasting. Site-scale models like TECO-CNP can fully leverage rich, localized datasets, including forest inventory records, experimental manipulations, and eddy covariance measurements, to constrain model parameters and processes. This integration is crucial because unobserved or weakly observed processes cannot be reliably constrained through data assimilation alone (Luo et al., 2011). TECO-CNP is designed to facilitate the fusion of such multi-process information, thereby enabling more mechanistic and robust representations of ecosystem C-N-P dynamics. In contrast, global-scale models often rely on aggregated or remote-sensing data and typically lack the resolution or flexibility to assimilate detailed, site-specific measurements. By bridging observational data and predictive capacity at the process level, TECO-CNP provides a powerful tool for advancing both model accuracy and ecological understanding at ecosystem-relevant scales. We have emphasized these features more clearly in the revised manuscript (L773-777; L949-958).

**Comment 4B:** *How were the soil P pools at the Tiantong site measured? Did you compare the measured soil P pool with other studies? They seem lower than other studies (Fig. 5c).*

**Response:** Soil phosphorus (P) pools at the Tiantong site were measured using systematic sampling across 185 grid points (each 20 × 20 m) within the permanent Tiantong forest plot (a member site of ForestGEO; https://forestgeo.si.edu; Fig. 3). At each grid point, soil samples were collected at three depth intervals (0-20, 20-40, 40-60 cm) using a 5 cm diameter auger, with three replicates per depth.

The measured soil P pool for the 0-60 cm profile was $181.59 \pm 60.18$ g P m$^{-2}$ (Table S3), which is well captured by the TECO-CNP simulation (157.7 g P m$^{-2}$; Fig. 5c). These values fall within the typical range reported for forest ecosystems in China (220.15, interquartile range: 130.74-341.98 g P m$^{-2}$; Zhu et al., 2020), suggesting our field-based estimates are robust.

Please note that some values in Figure 5c were rescaled for visualization purposes. To avoid confusion, we have now added an explanatory note to the figure caption (L789-790). Further details on the soil sampling methodology have also been provided in the revised text (L625-639).

**Comment 5B:** *The simulation results of C-only, CN, and CNP versions were compared in this study to prove the good performance of the CNP model. Did you calibrate these three versions individually? Was the same parameter optimization algorithm applied to all three versions? Or you just calibrate the only CNP model, and apply the same parameters to other versions. Did you simulate the C cycle in this site by using C-only or CN versions before the development of CNP? How well did these two models perform? Many parameters are constant values, such as Vre. I guess these parameters were not calibrated but derived from the literature. Were these parameters suitable for subtropical forest ecosystems?*

**Response:** Thank you for these detailed questions. In this revised version, we have provided the following clarifications regarding model calibration, parameter sources, and the rationale for using a CNP model:

**(1)** *Model calibration and parameterization strategy*: All three model versions (C-only, CN, and CNP) were calibrated using the same set of site-specific parameters derived from a measurement-informed approach. Key parameters, including specific leaf area (SLA), $V_{cmax}$, $J_{max}$, plant height, nutrient resorption fractions, and stoichiometric ratios, were obtained from in situ field measurements at the Tiantong site (Section 2.3.2; Tables S1-S3). This unified calibration ensures that differences in model performance reflect structural differences rather than inconsistencies in parameterization.

For parameters not directly measurable on-site, such as occluded P release rate ($v_{re}$), as noted by the reviewer, we used values informed by experimental studies and calibrated land surface models appropriate for the vegetation type (Table 4). For instance, the classification of Tiantong soils as *Ultisols* (Song & Wang, 1995) informed our selection of parameters for P weathering and sorption (e.g., $K_s$ and $S_{max}$), while the subtropical evergreen broadleaf forest context guided the parameterization of P mineralization and allocation processes (e.g., Wang et al., 2010; Arora & Boer, 2005). To ensure ecological relevance, all literature-derived parameters were carefully evaluated against site-specific characteristics. This combined approach preserves mechanistic realism while incorporating the best available knowledge. We have clarified this approach in the revised text at L291-294 and L602-624

**(2)** *Model independence and comparison procedure*: Although the C-only model served as the structural foundation for model development, the C-only, CN, and CNP versions were implemented as independent configurations. Each version was run separately after spin-up using

the same calibrated parameter set, and no parameters were shared post-optimization. Model comparisons were based on simulations prior to data assimilation (Section 2.3.4), allowing an unbiased evaluation of structural differences (Fig. 5, Table 5). The results show that the CNP model achieved the closest agreement with observed carbon, nitrogen, and phosphorus pools and fluxes. We have made this finding clearer in this revised version (L744-746, L828-833, and L862-864).

**(3)** ***Data assimilation and parameter equifinality***: Only the CNP model was subjected to data assimilation in the original version of this study, as our focus was on evaluating the performance of the fully coupled structure. However, to address the reviewer's question, we conducted a supplementary analysis applying data assimilation to the C-only model. We found that the optimized C-only model achieved performance similar to the CNP model in terms of flux predictions. This similarity, however, stems from compensatory parameter behavior. For example, Fig. RB1 (Fig. S2 in revised supplementary materials) shows that the C-only model required SLA values far outside the observed community-level distribution, whereas the CNP model constrained parameter values consistent with field data. This reflects the well-documented issue of equifinality (Luo et al., 2016; Sierra et al., 2015), where simpler models can match outputs by adjusting unrelated parameters, thereby losing ecological interpretability. We have added sentences to make this point clearer in this version (L949-958 and Text S1).

[Figure]

**Figure S2**. Posterior distribution of constrained specific leaf area (SLA) and observed community-level SLA distribution. Green, blue, red, and gray represent C, CN, CNP models and observations, respectively. Vertical lines represent distribution means. D, Kolmogorov-Smirnov statistic.

**Comment 6B:** *The vegetation and soil pools simulated by the three versions were compared in section 3.1. What about C and nutrient fluxes? In Fig.7, I cannot identify the simulated NEE by the three model configurations.*

**Response:** Thank you for pointing this out. In the revised manuscript, carbon and nutrient fluxes are more clearly illustrated in the revised Table 5 and described in lines 758-761, 828-833 and 862-864 in Section 3.1. These additions illustrate the expected effects on carbon fluxes under phosphorus limitation in the CNP model, and the lower nitrogen mineralization due to reduced

litter input and constrained microbial activity. To further support transparency, we have included simulated nitrogen and phosphorus flux data in the supplementary repository for potential use in future comparative studies.

Regarding the original Figure 7, we have corrected the caption to clarify that it shows results from the CNP model before and after data assimilation, rather than from the three model versions. We apologize for the earlier misstatement and have updated the caption accordingly (Line 922).

**Table 5.** Observed and simulated carbon, nitrogen and phosphorus fluxes with C, CN and CNP configurations. The plant litterfall rate is the sum of litterfall of leaf, wood and reproductive pool.

| C, N and P fluxes | C-only | CN | CNP | Observation | Unit |
|---|---|---|---|---|---|
| C transfer from leaf to litter | 0.43 | 0.38 | 0.25 | 0.26±0.06 | kg C m$^{-2}$ yr$^{-1}$ |
| C transfer from plant to litter | 0.98 | 0.86 | 0.54 | 0.44±0.04 | kg C m$^{-2}$ yr$^{-1}$ |
| N transfer from plant to litter | - | 11.36 | 7.44 | 6.74±0.68 | g N m$^{-2}$ yr$^{-1}$ |
| P transfer from plant to litter | - | - | 0.24 | 0.79±0.24 | g P m$^{-2}$ yr$^{-1}$ |
| Soil respiration | 1.72 | 1.59 | 1.13 | 0.99±0.07 | kg C m$^{-2}$ yr$^{-1}$ |
| Net N mineralization | - | 18 | 12.3 | 13.14±0.73 | g N m$^{-2}$ yr$^{-1}$ |
| Net P mineralization | - | - | 0.54 | 0.67±0.14[a] | g P m$^{-2}$ yr$^{-1}$ |

[a]Jiang et al., 2024

**Minor comments**

**Comment 7B:** *L491-493. Please list the equations of P loss from SOM pools.*
**Response:** Done. We have added the Equation 44 to represent soil P loss from SOM pools at lines 481-484 in the revised manuscript.

**Comment 8B:** *Equation 54. What is the meaning of Pl*
**Response:** 'Pl' should be 'P$_{lab}$' representing the labile phosphorus pool. This has been corrected in the revised manuscript.

**Comment 9B:** *L607. Please correct the reference of Xu et al.*
**Response:** Corrected.

**Comment 10B:** *Fig 8. What is the meaning of the posterior distribution of parameters? Do they change with time?*
**Response:** In the revised version, we have explained the posterior distribution of parameters and its independence upon time (L675-677). The posterior parameter distribution represents our updated knowledge about parameter values after incorporating observational data through Bayesian inference, quantifying both the most likely parameter estimates and their associated uncertainties. In the context of our model, these distributions show which parameter values (e.g.,

$Q_{10}$, turnover times) are most consistent with observed carbon flux data and provide confidence intervals for those estimates.

Regarding temporal behavior, posterior distributions in our study do not change with time. We employed batch data assimilation (Evensen, 2009), processing the entire observational time series simultaneously to generate a single posterior distribution for each parameter. This approach assumes parameters are time-invariant ecological properties and yields static probability distributions representing the best parameter estimates constrained by all available observations. Sequential assimilation approaches that update posteriors over time are possible but were not employed in this study, as our objective was to characterize fixed ecosystem parameters rather than track temporal parameter evolution.

**Comment 11B:** *Table. How did you identify these target parameters? Did you conduct a sensitivity analysis?*

**Response:** Yes, we conducted a preliminary sensitivity analysis to support the selection of target parameters for data assimilation. As shown in Table RB2 (added to the revised supplementary materials as Table S6), parameters related to photosynthesis (SLA and $V_{cmax}$) and ecosystem respiration ($Q_{10}$) exhibited high sensitivity indices ($> 0.1$) with respect to GPP, ER, and NEE. These parameters were therefore selected for assimilation. In addition, we included all carbon pool turnover parameters ($T_1$–$T_9$) without pre-screening, as these govern carbon residence times and are crucial for matching observed pool dynamics, even if their sensitivity indices were lower. In summary, we selected assimilation parameters based on both sensitivity analysis and their direct link to observed ecological processes, rather than relying solely on pre-screening. We have made it clear in the revised version at lines 654-660.

**Table RB2.** Sensitivity index of selected parameters. $SI_{GPP}$, $SI_{ER}$, and $SI_{NEE}$ represent the sensitivity indices of gross primary productivity (GPP), ecosystem respiration (ER), and net ecosystem exchange (NEE) to each parameter, respectively. Bold values indicate sensitivity indices $> 0.1$.

| | $SI_{GPP}$ | $SI_{ER}$ | $SI_{NEE}$ |
|---|---|---|---|
| **Q10** | 0.009 | **0.554** | **0.750** |
| **SLA** | **0.112** | **0.110** | **0.115** |
| **Vcmax** | **0.820** | **0.210** | **1.669** |
| T1 | 0.023 | 0.027 | 0.091 |
| T2 | 0.000 | 0.036 | 0.050 |
| T3 | 0.008 | 0.025 | 0.055 |
| T4 | 0.000 | 0.026 | 0.037 |
| T5 | 0.004 | 0.039 | 0.065 |
| **T6** | 0.004 | **0.185** | **0.268** |
| **T7** | 0.008 | **0.239** | **0.313** |
| **T8** | 0.001 | **0.110** | **0.150** |
| T9 | 0.000 | 0.005 | 0.006 |

**Reference**

Arora, V. K., & Boer, G. J. (2005). A parameterization of leaf phenology for the terrestrial ecosystem component of climate models. Global Change Biology, 11, 39–59. https://doi.org/10.1111/j.1365-2486.2004.00890.x

Evensen, G. (2009). Data Assimilation: The Ensemble Kalman Filter. Springer-Verlag, Berlin Heidelberg, pp. 22-37.

Hartmann, H., Bahn, M., Carbone, M., & Richardson, A. D. (2020). Plant carbon allocation in a changing world – challenges and progress: Introduction to a Virtual Issue on carbon allocation. New Phytologist, 227(4), 981–988. https://doi.org/10.1111/nph.16757.

Haverd, V., Smith, B., Nieradzik, L., Briggs, P. R., Woodgate, W., Trudinger, C. M., Canadell, J. G., & Cuntz, M. (2018). A new version of the CABLE land surface model (Subversion revision r4601) incorporating land use and land cover change, woody vegetation demography, and a novel optimisation-based approach to plant coordination of photosynthesis. Geoscientific Model Development, 11, 2995–3026. https://doi.org/10.5194/gmd-11-2995-2018

Luo, Y., Ogle, K., Tucker, C., Fei, S., Gao, C., LaDeau, S., Clark, J. S., & Schimel, D. S. (2011). Ecological forecasting and data assimilation in a data-rich era. Ecological Applications, 21(5), 1429–1442. https://doi.org/10.1890/09-1275.1.

Merganičová, K., Merganič, J., Lehtonen, A., Vacchiano, G., Sever, M. Z. O., Augustynczik, A. L. D., Grote, R., Kyselová, I., Mäkelä, A., Yousefpour, R., Krejza, J., Collalti, A., & Reyer, C. P. O. (2019). Forest carbon allocation modelling under climate change. Tree Physiology, 39(12), 1937–1960. https://doi.org/10.1093/treephys/tpz105.

Nakhavali, M. A., Mercado, L. M., Hartley, I. P., Sitch, S., Cunha, F. V., Di Ponzio, R., & Camargo, J. L. (2022). Representation of the phosphorus cycle in the Joint UK Land Environment Simulator (vn5.5_JULES-CNP). Geoscientific Model Development, 15, 5241–5269. https://doi.org/10.5194/gmd-15-5241-2022.

Song, Y. C., & Wang, X. R. (1995). Vegetation and flora of Tiantong National Forest Park Zhejiang Province. Shanghai Scientific and Technical Document Publishing House.

Wang, Y. P., Law, R. M., & Pak, B. (2010). A global model of carbon, nitrogen and phosphorus cycles for the terrestrial biosphere. Biogeosciences, 7, 2261–2282. https://doi.org/10.5194/bg-7-2261-2010.

Zhu, J., Wu, A., & Zhou, G. (2021). Spatial distribution patterns of soil total phosphorus influenced by climatic factors in China's forest ecosystems. Scientific Reports, 11, 5357. https://doi.org/10.1038/s41598-021-84166-0.

Zhu, Q., Riley, W. J., Tang, J., Collier, N., Hoffman, F. M., Yang, X., & Bisht, G. (2019). Representing Nitrogen, Phosphorus, and Carbon Interactions in the E3SM Land Model: Development and Global Benchmarking. Journal of Advances in Modeling Earth Systems, 11(7), 2238–2258. https://doi.org/10.1029/2018MS001571.